# THE INFORMATION BOTTLENECK OF CHAIN-OF-THOUGHT AND HOW LATENT COT OVERCOMES IT

## ABSTRACT

Chain-of-thought (CoT) has become the de facto paradigm for large language models (LLMs) to solve complex reasoning tasks. However, due to the sequential nature of token generation, the inference time can be formidable if the CoT is exceedingly long. This paper identifies a fundamental *information bottleneck* that can cause the CoT to be long: although each forward pass can activate a vast amount of neurons, in the end, the information the model writes down is limited to a single token, making it inevitable to produce many more CoT steps than necessary. We first theoretically establish this bottleneck by showing that for some natural problems, such as pointer chasing and computing parity, either 1-layer transformers or constant-layer finite-precision transformers require a rather long CoT to solve. We then demonstrate that for these same problems, allowing the Transformer to write high-dimensional embeddings to the CoT (i.e., using latent CoT) significantly reduces the CoT length, establishing a provably theoretical benefit for using latent CoT. We further validate our theory with controlled experiments: training a small transformer to simulate Conway's Game of Life with latent CoT, we vary the per-step write bandwidth to the latent CoT and observe a sharp success threshold proportional to the board size.

## 1 INTRODUCTION

Chain-of-thought (CoT) reasoning has emerged as a powerful paradigm for large language models, which enables them to tackle complex reasoning tasks by decomposing them into intermediate steps before producing a final answer. However, since every token in CoT needs to be sequentially generated, the inference time of CoT grows linearly or even quadratically with the length of CoT.

Recently, a growing body of works has focused on reducing the length of CoT while maintaining the reasoning capability. Many studies incorporate length penalty designs into Reinforcement Learning (RL) (Kimi Team et al., 2025; Luo et al., 2025; Aggarwal & Welleck, 2025; Arora & Zanette, 2025; Gao et al., 2025). Others investigate prompting strategies that encourage LLMs to produce concise CoT in certain concise forms (Renze & Guven, 2024; Xu et al., 2025; Aytes et al., 2025), or fine-tuning approaches that train LLMs on compressed CoT samples (Kang et al., 2025; Xia et al., 2025; Cheng & Van Durme, 2024).

**Information Bottleneck in CoT.** *How much can the CoT be made shorter without sacrificing the reasoning capability of LLMs?* In this paper, we identify a fundamental limitation of all the above methods that cannot be overcome without changing the current CoT paradigm: the *information bottleneck* in CoT.

More specifically, each forward pass of a LLM only appends a single token to the transcript, which only conveys $O(\log |\mathcal{V}|)$ bits of information if the vocabulary size is $|\mathcal{V}|$. Therefore, in every decoding step, the model can only use $O(\log |\mathcal{V}|)$ more bits of information than the previous step, and write back $O(\log |\mathcal{V}|)$ bits of new information to the transcript. This slow accumulation of information forces the model to use many more CoT steps than necessary if the reasoning process needs a large amount of information to make progress. This $O(\log |\mathcal{V}|)$ bits constraint of the amount of new information is what we call *the information bottleneck.*

This information bottleneck would not be called a "bottleneck" if the model is indeed only able to produce $O(\log |\mathcal{V}|)$ bits of new information at each step. However, modern Transformer architectures

generate high-dimensional internal states at each forward pass. These rich hidden representations propagate layer by layer through residual streams, MLPs, and attention mechanisms. But at the final layer, they are abruptly compressed into a single token. This means that the model can "think" in a high-dimensional space with multiple layers of computation, but can only "write down" its thoughts through a narrow, low-bandwidth token, which limits the amount of information that can be passed to the next step.

**Latent CoT Overcomes the Information Bottleneck.**    Instead of appending a single token to the transcript at each step, allowing the model to append a high-dimensional embedding to the transcript at each step can overcome the information bottleneck and significantly reduce the CoT length. This strategy is commonly referred to as *latent CoT* (Hao et al. (2024); Zhu et al. (2025); Su et al. (2025); Shen et al. (2025)), which means each entry of CoT is not a token but a $d_{\text{model}}$ dimentional vector. In this work, we focus on the efficiency aspect of latent CoT. Using latent CoT, if properly designed to convey more information at each step, the CoT length can be significantly reduced.

## 1.1  OUR CONTRIBUTIONS

In this paper, we formalize the intuition above and demonstrate the information bottleneck in CoT with a series of theoretical results.

**Lower Bound for CoT Length.**    First, we show theoretical results for two classical problems: Pointer Chasing and Parity. For their definition, please refer to 2.

For both problems, we need a large number of token CoTs due to the information bottleneck.

**Theorem 1.1.** *The following holds:*

- *For a variant of the pointer chasing function (Definition 3.6), a single-layer transformer with dimension $d$ needs $\Omega(n/d)$ CoT steps.*

- *For the parity function, a constant-layer finite-precision transformer with $\mathsf{poly}(n)$ model dimension needs $\Omega(n/\mathsf{polylog}(n))$ CoT steps to solve.*

Remarkably, our lower bounds for the parity function hold regardless of the model dimension width and the computational cost of each forward pass per step.

**The Benefit of Latent CoT.**    Further, we show that if we use latent CoT, the CoT length can be significantly reduced.

**Theorem 1.2.** *The following holds:*

- *A single-layer transformer with dimension $d$ can solve the same variant of pointer chasing in $O(n/d^2+1)$ latent CoT steps.*

- *For the parity function, a constant-layer finite-precision transformer with $d$ model dimension can solve it in $O(n/d + \log n)$ steps with latent CoT.*

Both of the upper bounds with latent CoT improve roughly a factor of $d$ compared to the token CoT steps. This is close to the theoretical maximum improvement since one can always use roughly $O(d)$ tokens to record a $d$-dimensional embedding. Moreover, when $d$ is large enough (at least $\sqrt{n}$ in the first case and at least $n$ in the second case), latent CoT only needs $O(1)$ or $O(\log n)$ steps, while token CoT requires significantly more steps.

Taking our lower bounds and upper bounds together demonstrates that the inability of the transformers with token CoT to solve either pointer chasing or parity is not a lack of computational power—switching to latent CoT does not give the transformer any more computational bandwidth, but rather the inforamtion bottleneck as we identified—the only difference between latent CoT and token Cot is that now the transformer can write more information to the transcript.

**Experimental Verification.**    We validate the information–bandwidth view with a controlled study on Conway's Game of Life. We generate $n$ by $n$ ($n = 6, 8, 10$) boards with a random initial state

(alive/dead independently sampled for each cell). Then simulate it for $k = 10$ steps using a short CoT of the same length. The key experimental knob is the width of the `info_bottleneck` layer that gates what the model can communicate by CoT from one latent step to the next.

Roughly speaking, we have a tunable knob `info_bottleneck`, which corresponds to the dimensionality of the vectors in our short latent CoT. Interetstingly, this provides a unified view of different CoTs. When we set the bottleneck to $0$, each CoT output carrys no information, and this is the same as dot-by-dot CoT (Pfau et al. (2024)). When we set it to $\log_2(\text{vocab size})$, each CoT output carries the same amount of information as a single token, and this captures the usual token CoT. Finally, when we set the bottleneck to the model dimension $d_{\text{model}}$, it captures latent CoT (Hao et al. (2024)).

In our experiment, across $n = 6, 8, 10$, we observe a threshold in bottleneck (Figure 1). Below the threshold, test accuracy stays near chance and test loss remains high; above it, accuracy jumps to near-perfect while test loss drops steeply. This threshold grows as the problem complexity $(n^2)$ grows.

We can draw two conclusions: (i) The performance of token CoT suffers from its small information bottleneck; (ii) Note that we do not set any bottleneck on the information that is moving around by attention. This means that although attention is good at moving information around, it cannot replace the information that CoT passes from one latent step to another through the bottleneck.

These results are consistent with our theory: when the information bottleneck of CoT is too small, the model cannot propagate enough information through time and fails abruptly; once the bottleneck clears that threshold, the same architecture and training budget solve the task reliably. Latent CoT removes the discrete-token bottleneck and allows the transformer to efficiently utilize the short CoT.

## 1.2 Additional Related Works

**Theoretical Limitations of CoT.** Recent work has begun to probe the fundamental limits of CoT reasoning. Bavandpour et al. (2025) prove lower bounds on the length of CoT required for *Hard-attention Transformers* to solve certain reasoning tasks, such as Parity or Multiplication. A *Hard-attention Transformers* is a tranformer where each attention head can only attend to the unique position with maximum attention score. Similar to our result, they prove that parity requires $\Omega(n)$ length CoT, but only for *Hard-attention Transformers*, while our result holds for general transformers. The key insight to their result is how *Hard-attention Tranformers* simplifies under *random restictions*, a technique first applied to tranformers by Hahn & Rofin (2024). In contrast, our work identifies the information bottleneck of token CoT as a critical constraint, which is a completely different insight into the limitation of token CoTs.

**Token Complexity and Optimal Length.** Our theoretical analysis provides a formal grounding for recent empirical observations regarding CoT efficiency. The "Token Complexity Hypothesis" (Lee et al., 2025) suggests that each task has its intrinsic *token complexity*, and LLMs struggle to compress their reasoning into fewer tokens than this complexity. Our results explain this by showing that the channel capacity of a single token is insufficient to carry complex state updates, necessitating a long chain of uncompressed tokens. Similarly, Yi et al. (2025) empirically identify a "sample optimal length" for inference; our work theoretically justifies why this length cannot be arbitrarily shortened with discrete tokens but can be significantly reduced with latent embeddings.

**Information Bottlenecks.** The concept of information bottlenecks in LLMs has also been explored in the context of the attention mechanism itself. Schnabel et al. (2025) argue that information is "lost in transmission" across attention layers, hindering global reasoning. Their bottleck is mainly for transformers without CoT, and they additionally showed that CoT can break their bottleneck. Our work highlights a distinct bottleneck: the token CoT itself. Even if the internal attention mechanism preserves global information, and even if with token CoT, the requirement to output a single token forces a lossy compression at every step.

**Latent CoT and Looped Transformers.** Finally, our proposed solution aligns with the growing interest in continuous reasoning and latent CoT (Hao et al., 2024; Zhu et al., 2025), demonstrating its advantages in practice.

It worth mentioning that in the COCONUT paper (Hao et al., 2024), they observes that in their experiment, latent CoT outperforms token CoT on logical reasoning datasets like ProntoQA and the authors' new dataset ProsQA. They argue this is because COCONUT's continuous latent state can encode multiple potential next steps simultaneously.This is exactly one of the capabilities that requires a high-capacity information channel. Our paper provides a theoretical explanation and a more general view for these mixed results: For tasks with **low state-passing needs** (e.g., simple reasoning), the $O(\log |\mathcal{V}|)$ bottleneck of token CoT is *sufficient*; For tasks with **high state-passing needs** (like our PARITY or GoL), the $O(\log |\mathcal{V}|)$ bottleneck is *insufficient*. Here, latent CoT shows a massive performance gain, as predicted by our theory.

The other related line of work is the study of Looped Transformers (Saunshi et al., 2025; Geiping et al., 2025; Chen et al., 2025; Yang et al., 2023; Li et al., 2025; Eyuboglu et al., 2024; Xu & Sato, 2025). Expressibility-wise, They are equivalent to an internal latent CoT of the model. In particular, (Xu & Sato, 2025) shows that in terms of abstract complexity class, looped transformer (same holds for latent CoT) with $\log^k n$ steps is a larger class than token CoT of same number of steps. In comparison, our work not only shows *conrete problems* that seperate the two but also indentifies *information bottleneck* as the fundamental reason behind.

## 2 PRELIMINARY

Throughout the paper, we use $n$ to denote the maximum prompt length, $d$ the model dimension. We consider a decoder-only Transformer architecture and the detailed description can be found at Appendix A.1.

**Definitions of classical problems.** We first present a few definition of classical problems studied in our paper.

1. **Pointer Chasing.** Given two functions $f_A, f_B \colon [m] \to [m]$ and an integer $k$, find out $(f_B \circ f_A)^{(k)}(1)$. Here $f_B \circ f_A$ denotes function composition, and the superscript $(k)$ denote composing the underlying function with itself $k$ times.

2. **Parity.** Given a sequence of $n$ tokens $x_1, \ldots, x_n$, each of which is either 0 or 1, compute the parity of the sequence, i.e., $x_1 \oplus \cdots \oplus x_n$.

3. **Conway's Game of Life.** Given an initial board $S^{(0)} \in \{0,1\}^{n \times n}$ (1 = alive, 0 = dead) and an integer $k$, iteratively compute $S^{(t)}$ for $t = 1, \ldots, k$ under the standard Life rule: a cell is alive in step $t$ iff it had exactly three live neighbors at step $t-1$, or it had two live neighbors and was already alive. The task is to output the final configuration $S^{(k)}$ (or any statistic derived from it, such as the number of live cells).

In all three problems, the prompt is provided as a tokenized textual description (e.g., enumerating function tables, the bit string, or the initial board), but our results depends only on the underlying information content and therefore not on the specific input encoding scheme.

**Boolean function analysis** We present some basic terminology for boolean function analysis. Let $f : \{-1,1\}^n \to \mathbb{R}$ be a Boolean function. Its Fourier expansion is given by

$$f(x) = \sum_{S \subseteq [n]} \hat{f}(S) \chi_S(x),$$

where $\chi_S(x) = \prod_{i \in S} x_i$ is the character function and

$$\hat{f}(S) = \mathbb{E}_{x \sim \{-1,1\}^n}[f(x)\chi_S(x)]$$

is the Fourier coefficient corresponding to subset $S \subseteq [n]$. When the output domain is $\{-1,1\}$, one further have

$$\sum_{S \subseteq [n]} \hat{f}(S)^2 = 1 \qquad \text{(Parseval's Identity)} \tag{1}$$

**Definition 2.1** (Sensitivity). *Let* $f : \{-1, 1\}^n \to \{-1, 1\}$ *be a Boolean function. The average sensitivity of* $f$, *denoted* $AS(f)$, *is defined as:*

$$AS(f) = \sum_{i=1}^{n} \Pr_{x \sim \{-1,1\}^n} [f(x) \neq f(x^{\oplus i})],$$

*where* $x^{\oplus i}$ *is the vector* $x$ *with the* $i$-*th bit flipped. Alternatively, using the Fourier expansion of* $f$, *we have:*

$$AS(f) = 4 \sum_{S \subseteq [n]} |S| \cdot \hat{f}(S)^2.$$

For further background on Boolean function analysis, please refer to the classical textbook by O'Donnell (2014).

**Communication Complexity.** Communication complexity studies the number of bits that distributed parties must exchange to jointly compute a function of their distributed inputs. We rely on standard definitions for randomized protocols with *public randomness*.

- **Two-party Randomized Protocol.** Alice holds input $x \in \mathcal{X}$ and Bob holds input $y \in \mathcal{Y}$. They share a public random string $r$. The protocol proceeds in rounds where parties exchange messages depending on their private input, the public randomness, and prior messages. The *communication cost* is the maximum total number of bits transmitted over all inputs and random strings. The *round complexity* is the number of message exchanges.

- **Success Probability and Advantage.** A protocol computes a Boolean function $f(x, y)$ with success probability $1/2 + \eta$ if and only if $\Pr[\text{output} = f(x, y)] \geq 1/2 + \eta$ for every input pair $(x, y)$. We call $\eta \in [0, 1/2]$ the *advantage*. We say a protocol has nontrivial constant success probability if $\eta = \Theta(1)$.

- **Direct Sum (XOR Lemma).** For a Boolean function $f$, let $f^{\oplus s}(x^{(1)}, y^{(1)}, \dots, x^{(s)}, y^{(s)}) \triangleq \bigoplus_{i=1}^{s} f(x^{(i)}, y^{(i)})$ denote the XOR of $s$ independent instances. Strong XOR lemmas (e.g., Yu (2022)) state that if any $r$-round protocol for $f$ with constant success probability requires communication cost $C$, then computing $f^{\oplus s}$ with advantage $2^{-O(s)}$ requires cost roughly $\Omega(s \cdot C)$.

- **Laconic (Three-party) Communication.** To model the information bottleneck in token-based CoT, we introduce a three-party model with Alice, Bob, and a central coordinator Charlie. In each round $t$:
  1. Charlie broadcasts the current "token" $z_{t-1} \in \{0, 1\}^p$ (where $p$ is small, e.g., $O(\log n)$) to Alice and Bob.
  2. Alice and Bob send high-dimensional messages $m_A, m_B \in \{0, 1\}^{O(dp)}$ to Charlie based on their inputs and $z_{t-1}$.
  3. Charlie computes the next token $z_t$ from $(m_A, m_B, z_{t-1})$ and discards $m_A, m_B$.

  Crucially, only the low-bandwidth token $z_t$ persists to the next round. This mirrors a Transformer where high-dimensional internal activations $(m_A, m_B)$ are compressed into a single output token $(z_t)$. This is the *information bottleneck*.

## 3 SEPARATIONS FOR ONE-LAYER TRANSFORMER

**Theorem 3.1.** *Let* $n$ *be the maximum prompt length,* $d$ *be the model dimension, all arithmetic operations are performed with* $p = O(\log(n))$ *bits of precisions and the vocabulary size is of* $|\mathcal{V}| = \text{poly}(n)$. *For 1-layer Transformer, there is a task such that*

- *it requires at least* $n_{\text{cot}} = \Omega(n/d \log(n))$ *CoT steps, while*

- *it can be solved with* $O(n/d^2 + 1)$ *latent CoT steps.*

We sketch the proof idea of the lower bound for CoT, the missing details can be found at Appendix C.

We propose a fine grained communication model that not only captures the information bottleck of an attention layer, but also the information bottleneck of token representation.

**Definition 3.2** (Laconic communication model). *Consider the following three-party communication model. Alice and Bob each hold a private input $z_A$ and $z_B$ $((z_A, z_B) \in \{0,1\}^{np})$ and Charlie initially holds the empty string. They wish to collectively compute some function value $f(z_A, z_B) \in \{0,1\}$. The communciation proceeds in a sequence of $R$ rounds and at round $r = 1, 2, \ldots, R$,*

- *Charlie sends its input $z_{C,<r} \in \{0,1\}^{(r-1)p}$ to Alice and Bob*

- *Alice (resp. Bob) then replies with $\Pi_{A,r} \in \{0,1\}^{2dp}$ (resp. $\Pi_{B,r}$) based on $z_{C,<r}$ and its own input $z_A$ (resp. $z_B$).*

- *Given $z_{C,<r} \in \{0,1\}^{(r-1)p}$ and the transcript $\Pi_{A,r}, \Pi_{B,r}$, Charlie compresses them into $z_r \in \{0,1\}^p$ and then augments $z_{C,<r} \in \{0,1\}^{(r-1)p}$ to $z_{C,<r+1} = (z_{C,<r}, z_r) \in \{0,1\}^{(r-1)p} \times \{0,1\}^p$.*

*Note the information of $\Pi_{A,r}, \Pi_{B,r}$ has been forgot except those in $z_r$. At the end, Charlie outputs the answer of $f(z_A, z_B)$.*

Intuitively, this model catpures a key feature of token-based CoT: after a lot of computation (resp. communication), only a token (resp. logarithmic bits) is written to the CoT (transcript).

We first prove that lower bounds in this model can be translated to CoT lower bounds against 1-layer transformers. We say that a Transformer solves the task in Definition 3.2 within $n_{\text{cot}}$ steps, if given $z_A, z_B$ as prompt, it correctly outputs the value of $f(z_A, z_B)$

**Lemma 3.3** (Reduction). *If there is a 1-layer Transformer that solves the task in Definition 3.2 after $n_{\text{cot}}$ steps, then there is a $(n_{\text{cot}} + 1)$-round communication protocol.*

It remains to present a hard function $f$ for the laconic communication model. We first introduce the pointer chasing task, which is a classic problem that has been extensively studied in the communication complexity literature (Papadimitriou & Sipser, 1982; Nisan & Widgerson, 1991; Klauck, 2000; Yehudayoff, 2020; Mao et al., 2024)

**Definition 3.4** (Pointer chasing). *In two-party pointer chasing problem, Alice and Bob each holds a function $f_A, f_B : [m] \to [m]$. Given $k \in [m]$, they are asked to compute the parity of the $\text{PC}_k(f_A, f_B) = (f_B \circ f_A)^{(k)}(1)$.*

We make use of the following communication lower bound.

**Lemma 3.5** (Lower bound for pointer chasing Mao et al. (2024)). *For any $k \in [m]$ and $(2k-1)$-round communcation protocol that exchanges 1 bit per round and that succeeds with probability at least $2/3$ over the uniform distribution, its communication complexity is $\Omega(m/k + k)$.*

In the proof, we would take $k = 2$ and $m = d$. The actual hard function $f$ is the XOR of $n/m$ pointer chasing instance.

**Definition 3.6** (XOR pointer chasing). *Alice and Bob each holds $n/2d$ functions $f_{A,i} : [d] \to [d]$ $(i \in [n/d])$ and $f_{B,i} : [d] \to [d]$ $(i \in [n/d])$. They wish the compute the XOR of these pointer chasing, that is $\bigoplus_{i \in [n/d]} \text{PC}_2(f_{A,i}, f_{B,i})$*

We need to following XOR Lemma for bounded round communication protocol.

**Lemma 3.7** (Strong XOR Lemma for bounded round communication Yu (2022)). *Suppose the communication complexity of an $\{0,1\}$-valued function $f$ is $C$ within $r$-round of communication (with success prob $2/3$), then the randomized communication complexity of computing $f^{\oplus s}$ with advantage $1/2 + 2^{-s}$ is at least $s \cdot (r^{-O(r)} \cdot C - 1)$*

We have the following lower bound for XOR pointer chasing.

**Lemma 3.8** (Lower bound for XOR pointer chasing). *For any 3-round communication protocol that solves the XOR pointer chasing task (Definition 3.6) with advantage $\frac{1}{2} + 2^{-n/2d}$, its communication complexity is at least $\Omega(n)$.*

Finally, we prove a lower bound for XOR pointer chasing under the laconic communication model using Lemma 3.7, this is the key step of our proof.

**Lemma 3.9** (Lower bound for laconic communcation). *The number of communication rounds to solve the XOR pointer chasing in the laconic communication is $\Omega(n/dp)$.*

Combining Lemma 3.3 and Lemma 3.9, we have proved the CoT lower bound in Theorem 3.1. The construction for the latent CoT is formally proved at Appendix C.

## 4 SEPERATION FOR MULTI-LAYER FINITE PRECISION TRANSFORMER

In this section, we present our lower bound results on the limitation of token CoT for constant depth,finite precision transformers, and contrast it with an upper bound for latent CoT. It is worth noting that this limitation cannot be overcome simply by increasing the number of Transformer layers. As the input size (and thus context length) grows, a fixed-depth cannot asymptotically match this growing demand.

**Theorem 4.1.** *Let $n$ be the maximum prompt length, $d$ be the model dimension. Consider the task of PARITY, for a constant depth, finite precision Transformer*

- *it needs at least $n_{\text{cot}} = \widetilde{\Omega}(n)$ CoT steps to solve PARITY[1]; while*

- *latent CoT requires only $O(n/d + \log n)$ steps to solve PARITY.*

We now present the proof for the CoT lower bound in Theorem 4.1, the construction for latent CoT is formally stated and proved as Theorem B.4.

Indeed, we prove a stronger result, which characterizes the representation power of constant depth finite precision Transformer using Fourier analysis.

**Theorem 4.2.** *Let $n$ be the input length. A constant depth, finite precision Transformer with model dimension $d = \mathsf{poly}(n)$ and $n_{\text{cot}}$ CoT steps have at most $|\mathcal{V}|^{2n_{\text{cot}}} \cdot 2^{-k/\mathsf{polylog}(n)}$ Fourier mass at level $k$ or above.*

Theorem 4.2 shows that a finite-precision Transformer can be approximated by a low-degree polynomial. After $n_{\text{cot}}$ steps, its Fourier mass is concentrated mostly on levels below $\widetilde{O}(n_{\text{cot}})$. There are many natural functions that have non-trivial mass on high degree coefficient, e.g., PARITY is the degree $n$ polynomial. This also include any functions with large average sensitivity.

**Corollary 4.3.** *Let $f$ be a function of average sensivity $AS(f)$ (see Definition 2.1), then constant depth finite precision Transformer requires $n_{\text{cot}} = \widetilde{\Omega}(AS(f))$ CoT steps to compute $f$.*

The corollary follows directly from Theorem 4.2, Definition 2.1 and Parseval's Identity (1).

In the rest part, our goal is to prove Theorem 4.2. First, we note that, without CoT, a finite precision Transformer can be simulated by a constant depth boolean circuit.

**Theorem 4.4** (Li et al. (2024)). *A finite precision, constant depth Transformer with model dimension $d = \mathsf{poly}(n)$ can be simulated by a constant depth, polynomial size boolean circuit.*

We have the following bounds on the Fourier spectral of polynomial size boolean circuits.

**Theorem 4.5** (Tal (2017); Håstad (2014)). *Let $f$ be a Boolean circuit with depth $L$ and size $m$. Then,*

$$\sum_{S:|S|\geq k} \hat{f}(S)^2 \leq 2 \cdot 2^{-k/O((\log m)^{L-1})}.$$

*Proof of Theorm 4.2.* Given a finite precision constant depth Transformer $\Gamma$, let $g : \{0,1\}^n \to \{0,1\}$ be the function computed by $\Gamma$ in $n_{\text{cot}}$ steps. For each $i \in [n_{\text{cot}}]$, let $g_i : \{0,1\}^n \times \mathcal{V}^{(i-1)} \to \mathcal{V}$ denote the function computed by $\Gamma$ (without CoT) on input sequences of length $n + i - 1$ under greedy decoding. Without loss of generality, we assume that $g(x) = 1$ if and only if the last generated token equals 1.

---

[1]this holds for any $d = \mathsf{poly}(n)$

For any $t \in \mathcal{V}^{n_{\text{cot}}}$, define $h_t : \{0,1\}^n \to \{0,1\}$ as:

$$h_t(x) = 1\{t_{n_{\text{cot}}} = 1\} \cdot \prod_{i \in [n_{\text{cot}}]} 1\{g_i(x, t_{<i}) = t_i\}$$

That is, $h_t(x)$ equals $g(x)$ if and only if $t$ is the sequence of CoT tokens generated by $\Gamma$ under greedy decoding; otherwise, $h_t(x) = 0$.

By Theorem 4.4, each indicator function within the definition of $h_t(x)$ corresponds to a constant-depth polynomial-size Boolean circuit. Hence, $h_t(x)$ itself is a constant-depth polynomial-size circuit (it can writen as the OR of these indicator functions).

We can express $g(x)$ as:

$$g(x) = \sum_{t \in \mathcal{V}^{n_{\text{cot}}}} h_t(x). \tag{2}$$

Transitioning to the $\{-1, 1\}$ basis, define $f, \{f_t\}_{t \in \mathcal{V}^{n_{\text{cot}}}} : \{-1, 1\}^n \to \{-1, 1\}$ as:

$$f(x) = 1 - 2g\left(\frac{1-x_1}{2}, \ldots, \frac{1-x_n}{2}\right), \, f_t(x) = 1 - 2h_t\left(\frac{1-x_1}{2}, \ldots, \frac{1-x_n}{2}\right), \quad \forall t \in \mathcal{V}^{n_{\text{cot}}}.$$

Then, by Eq. (2), we have:

$$\frac{1 - f(x)}{2} = \sum_{t \in \mathcal{V}^{n_{\text{cot}}}} \frac{1 - f_t(x)}{2}.$$

Thus, for any non-empty set $S$, the Fourier coefficients satisfy

$$\hat{f}(S) = \sum_{t \in \mathcal{V}^{n_{\text{cot}}}} \hat{f}_t(S).$$

Applying the Cauchy–Schwarz inequality yields:

$$\hat{f}(S)^2 \leq |\mathcal{V}|^{n_{\text{cot}}} \sum_{t \in \mathcal{V}^{n_{\text{cot}}}} \hat{f}_t(S)^2.$$

Consequently, by Theorem 4.5, we derive

$$\sum_{S:|S| \geq k} \hat{f}(S)^2 \leq |\mathcal{V}|^{n_{\text{cot}}} \sum_{t \in \mathcal{V}^{n_{\text{cot}}}} \sum_{S:|S| \geq k} \hat{f}_t(S)^2 \leq |\mathcal{V}|^{2n_{\text{cot}}} \cdot 2^{-k/\text{polylog}(n)}.$$

This compltes the proof. □

## 5 EXPERIMENTS

To empirically validate our hypothesis regarding the information bottleneck, we designed a controlled experiment using Conway's Game of Life. Our goal is to demonstrate that for such an iterative task, the performance of a model is critically dependent on the bandwidth of information passed between autoregressive steps. By employing a Latent Chain-of-Thought (Latent CoT) architecture, we directly manipulate this bandwidth and observe its effect on task performance.

### 5.1 EXPERIMENTAL SETUP

**Task: Conway's Game of Life**  We select Conway's Game of Life as our experimental testbed. The task is structured as follows: the model is presented with an initial $n \times n$ board state and is tasked with simulating the game for $k = 10$ steps. The final objective is to output the total number of live cells in the terminal configuration. We conduct experiments across three levels of complexity by varying the board size, with $n \in \{6, 8, 10\}$.

**Model Architecture: Latent CoT Transformer with information bottleneck** Our model is built upon a decoder-only transformer architecture with 5 decoder layers and we set $d_{\text{model}} = 512, d_{\text{itermediate}} = 2048$ and number of attention heads $H = 4$. It uses the `Qwen3` tokenizer and acts the same as the standard transformer architecture when prefilling the input prompt.

To control the information bandwidth in CoT, instead of generating textual tokens for CoT, our model performs $k$ latent reasoning steps internally. The architecture is as follows:

1. **`mlp1` (Encoder)**: This encodes the final hidden state into a latent representation, which represents the model's internal "thought" or the predicted state of the Game of Life board.

2. **`mlp2` (Decoder/Injector)**: This latent state is then processed by `mlp2` and injected back into the model as an input embedding to initiate the subsequent latent reasoning step. This process is iterated for $k = 10$ steps.

Crucially, `mlp2` is designed with an **information bottleneck** layer at last, implemented as the sequential composition of `nn.Linear(hidden_size, info_bottleneck` and `nn.Linear(info_bottleneck, hidden_size)`. The `info_bottleneck` dimension is the key hyperparameter we vary in our experiments, allowing us to directly control the informational capacity of the channel between latent reasoning steps.

We note that this setup gives a natural interpolation between a few CoT concepts:

- When `info_bottleneck` $= 0$, it captures the dot-by-dot CoT (Pfau et al. (2024));
- When `info_bottleneck` $= \log_2(\texttt{vocab\_size})$, this architecture captures the information passed by usual token CoT;
- When `info_bottleneck` $= d_{\text{model}}$, it captures lacent CoT (Hao et al. (2024)).

**Dataset and Representation** The training data is generated procedurally, and we train the internal reasoning steps with teacher forcing. For each sample, a random initial board is created. The ground-truth sequence of board states for $k = 10$ steps is pre-computed using the canonical Game of Life rules. Each board state is flattened and transformed into a one-hot encoded vector of size $n \times n \times 2$, representing the alive/dead status of each cell. This sequence of vectors $\mathbf{z}_1, \mathbf{z}_2, \ldots, \mathbf{z}_k$ serves as the ground-truth targets for our latent states. The prompt is formatted as a natural language question specifying the initial board and the task, followed by the ground-truth sequence of board states as CoT, and then the final answer is a string containing the number of live cells (e.g., "Answer: 23").

**Adding Noise to the Input** Although teacher forcing lets the model learns very efficiently how to predict the next reasoning step in a prallel forward pass, this was not enough because in the sequential latent reasoning steps, the model's prediction error accumulates and eventually overwhelms. To get around this, we add clamped Gaussian noise to the latent reasoning steps $\tilde{\mathbf{z}} = \mathbf{z} + \text{clamp}(\epsilon, -0.5, 0.5)$ where $\epsilon \sim \mathcal{N}(0, 0.3 \cdot \mathbf{I})$, so that the models learns to error correct its own prediction error.

## 5.2 TRAINING DETAILS

**Objective Function** Our training objective is a composite loss designed to supervise both the intermediate reasoning process and the final answer. The total loss $\mathcal{L}$ is the sum of a latent state loss $\mathcal{L}_{\text{latent}}$ and a final answer loss $\mathcal{L}_{\text{final}}$:

$$\mathcal{L} = \mathcal{L}_{\text{latent}} + \mathcal{L}_{\text{final}}$$

- **Latent Loss ($\mathcal{L}_{\text{latent}}$)**: To ensure the internal reasoning steps correspond to the actual game dynamics, we apply a supervision signal at each of the $k$ latent steps. The latent loss is the Mean Squared Error (L2 loss) between the model's generated latent state and the ground-truth one-hot encoded board state for that step.

$$\mathcal{L}_{\text{latent}} = \frac{1}{k} \sum_{i=1}^{k} \|S_{\text{pred}}^{(i)} - S_{\text{true}}^{(i)}\|_2^2$$

where $S^{(i)}$ is the latent state at step $i$.

- **Final Loss** ($\mathcal{L}_{\textbf{final}}$): To train the model to produce the correct final answer, we use a standard cross-entropy loss on the model's output logits, comparing the predicted token distribution against the ground-truth answer tokens.

**Hyperparameters and Optimization**    The model was trained using the AdamW optimizer with a learning rate of $2 \times 10^{-4}$ and weight decay of $0.01$. We employed a cosine learning rate schedule with a warmup ratio of $0.02$.

### 5.3 RESULTS AND ANALYSIS

**Training and Testing.**    During training, we use teacher forcing, meaning that we feed the transformer natural-language prompt together with a *perfect* latent CoT supervision constructed from the ground-truth Game-of-Life roll-out of $k = 10$ steps.

During testing (inference), we provide only the prompt; the model then rolls out $k = 10$ latent reasoning steps *autoregressively* through the information-bottleneck module (of width `info_bottleneck`) without access to ground-truth latents, and finally decodes the answer from the last hidden state. The test loss is calculated as $\mathcal{L} = \mathcal{L}_{\text{latent}} + \mathcal{L}_{\text{final}}$ as well. The test accuracy is calculated solely based on whether the final answer is the correct final number of living cells. We ablate the bottleneck width to study its effect on test loss and accuracy.

**Impact of the Information Bottleneck**    The state of an $n \times n$ board requires $n^2$ bits of information to be perfectly represented. We hypothesize that the `info_bottleneck` dimension must be large enough to accommodate this information. As illustrated in Figure 1, our results reveal a sharp phase transition: models with bottleneck dimensions below the threshold fail almost completely, while those above the threshold achieve high accuracy and low test loss. The critical threshold is related to the problem complexity ($n^2$), and increases as $n$ increases.

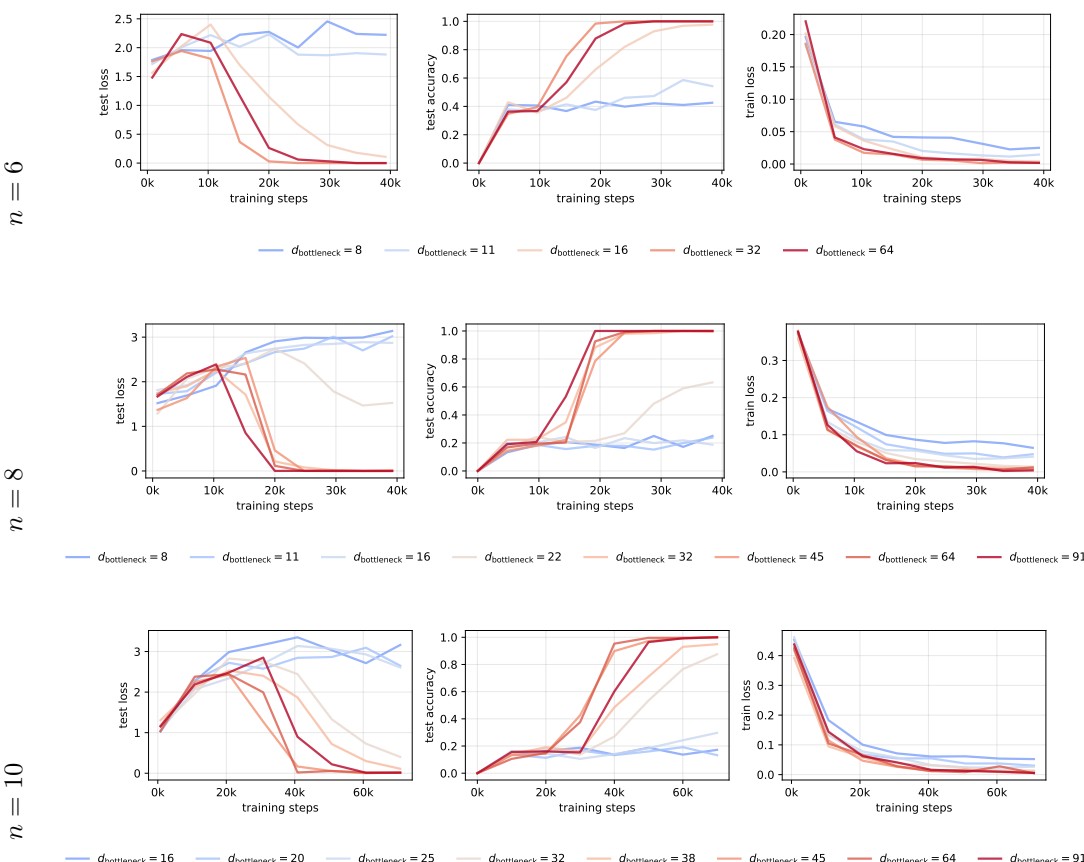

Figure 1: Results across different board sizes ($n = 6, 8, 10$) for $k = 10$ steps.

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

## A  NOTATIONS AND PRELIMINARIES

Our notations and mathematical definitions of transformers follows closely from Li et al. (2024). For completeness, we list them below.

We denote by $\mathbb{N}$ and $\mathbb{R}$ the sets of natural and real numbers, respectively. For any positive integer $n$, we write $[n] = \{1, 2, \ldots, n\}$. We define the ReLU activation function as $\mathsf{relu}(x) = \max(x, 0)$. For a vector $x$, we use $x_{a:b}$ to represent the subvector containing elements from position $a$ to position $b$. For a matrix $M$, we write $M_{a_1:b_1, a_2:b_2}$ to denote the submatrix formed by selecting rows from $a_1$ to $b_1$ and columns from $a_2$ to $b_2$. We also use $a_1 :$ to represent indices from $a_1$ to the end, $: b_1$ for indices from the beginning (1) to $b_1$, and $:$ for all indices.

We use $\phi(x) = \sum_{i=1}^{|x|} 2^{|x|-i} x_i$ to represent the decimal value of a binary string $x$. We denote by $\mathsf{bin}_k(x)$ the standard binary encoding of natural number $x$ using $k$ bits such that $\phi(\mathsf{bin}_k(x)) = x$, and by $\mathsf{sbin}_k(x)$ the signed binary encoding, defined as $2\mathsf{bin}_k(x) - (1, \ldots, 1)$. For any positive integer $n$, we define the softmax function $\mathrm{softmax} : \mathbb{R}^n \to \mathbb{R}^n$ as $(\mathrm{softmax}(x))_i = \exp(x_i)/\sum_{j=1}^{n} \exp(x_j)$ for any $x \in \mathbb{R}^n$ and $i \in [n]$. We use $\odot$ for element-wise multiplication of vectors. We denote vector concatenation by $a\|b$ or $(a, b)$.

### A.1  DECODER-ONLY TRANSFORMERS

Given a vocabulary $\mathcal{V}$, a *decoder-only* transformer with parameters $\theta$ and maximum input length $n_{\max}$ maps an input sequence $(x_1, \ldots, x_n) \in \mathcal{V}^n$ to a probability distribution over $\mathcal{V}$ for all $n \leq n_{\max}$, which we denote as $p_\theta(\cdot \mid x_1, \ldots, x_n)$. We also define $\mathsf{TF}_\theta(x)$ as the token in $\mathcal{V}$ that maximizes this probability distribution: $\mathsf{TF}_\theta(x_1, \ldots, x_n) = \arg\max_{y \in \mathcal{V}} p_\theta(y \mid x_1, \ldots, x_n)$.

**Next-token Generator:**   Given a vocabulary $\mathcal{V}$, a next-token generator with parameters $\theta$ and maximum input length $n_{\max}$ is a function from $\cup_{n=1}^{n_{\max}} \mathcal{V}^n$ to $\mathcal{V}$. The primary next-token generator we study is the decoder-only transformer $\mathsf{TF}_\theta(x_1, \ldots, x_n)$ where each $x_i \in \mathcal{V}$ for $i \in [n]$. We recursively define $\mathsf{TF}_\theta^i(x_1, \ldots, x_n) = \mathsf{TF}_\theta^{i-1}(x_1, \ldots, x_n, \mathsf{TF}_\theta(x_1, \ldots, x_n))$ for any positive integer $i$ and $n$ such that $i + n \leq n_{\max} - 1$, with base case $\mathsf{TF}_\theta^1(x_1, \ldots, x_n) = \mathsf{TF}_\theta(x_1, \ldots, x_n)$. Thus, for all $0 \leq i \leq n_{\max} - n - 1$, the output after $i$ steps of chain-of-thought is $x_{n+i+1} = \mathsf{TF}_\theta^{i+1}(x_1, \ldots, x_n) = \mathsf{TF}_\theta(x_1, \ldots, x_n, x_{n+1}, \ldots, x_{n+i})$.

**Transformer Architecture Overview:**   The decoder-only transformer architecture we consider closely follows GPT-style models (Radford et al., 2019) and comprises four main components: a token embedding layer (TE), a positional encoding layer (PE), an output linear layer (OUTPUT), and a stack of $L$ identical decoder layers, where $L$ represents the model depth. Each decoder layer contains two sublayers: a multi-head self-attention mechanism (ATTN) and a position-wise feed-forward network (FF). Each component has its own trainable parameters indexed by the layer name and depth for attention and feed-forward layers. [2] We can decompose the model parameters $\theta$ as: $\theta = (\theta_{\mathsf{PE}}, \theta_{\mathsf{TE}}, \theta_{\mathsf{OUTPUT}}, \{\theta_{\mathsf{ATTN}}^{(l)}, \theta_{\mathsf{FF}}^{(l)}\}_{l=0}^{L-1})$, all of which are trainable. (See formal definition in Algorithm 3). Throughout this work, we use $d$ to denote the embedding dimension of the transformer.

**Self-Attention Mechanism:**   Given attention parameters $\theta_{\mathsf{ATTN}} = (W_Q, W_K, W_V, W_O) \in \mathbb{R}^{d \times d} \times \mathbb{R}^{d \times d} \times \mathbb{R}^{d \times d} \times \mathbb{R}^{d \times d}$, we define the masked attention layer for decoder-only transformers in Algorithm 1. Note that multi-head attention can be defined similarly and it does not change the expressive power of constant-depth decoder-only transformers.

---

**Algorithm 1** Causal Self-Attention, ATTN

---

**Require:** Parameter $\theta_{\mathsf{ATTN}} = (W_Q, W_K, W_V, W_O)$, Input embedding $h = (h_1, \ldots, h_n) \in \mathbb{R}^{nd}$.
**Ensure:** Output embedding $h' = (h'_1, \ldots, h'_n) = \mathsf{ATTN}_{\theta_{\mathsf{ATTN}}}(h_1, \ldots, h_n)$.
  1: $q_i = W_Q h_i, k_i = W_K h_i, v_i = W_V h_i, \forall i \in [n]$
  2: $s_i = \mathrm{softmax}(\langle q_i, k_1 \rangle, \ldots, \langle q_i, k_i \rangle) \| (0, \ldots, 0)$.
  3: $h'_i = W_O \sum_{j=1}^n (s_i)_j v_j$.

---

**Feed-Forward Network:**   Given feed-forward network parameters $\theta_{\mathsf{FF}} = (W_1, b_1, W_2, b_2) \in \mathbb{R}^{d \times d} \times \mathbb{R}^d \times \mathbb{R}^{d \times d} \times \mathbb{R}^d$, we define the feed-forward layer $\mathsf{FF}_{\theta_{\mathsf{FF}}} : \mathbb{R}^d \to \mathbb{R}^d$ as $\mathsf{FF}_{\theta_{\mathsf{FF}}}(h) = W_2 \mathrm{relu}(W_1 h + b_1) + b_2$.

**Token Embedding:**   Given token embedding parameters $\theta_{\mathsf{TE}} \in \mathbb{R}^{d \times |\mathcal{V}|}$, we define the token embedding layer by treating $\theta_{\mathsf{TE}}$ as a mapping from $\mathcal{V}$ to $\mathbb{R}^d$, so that for any $x \in \mathcal{V}$, the token embedding is $\theta_{\mathsf{TE}}(x)$.

**Position Encoding:**   Given positional encoding parameters $\theta_{\mathsf{PE}} \in \mathbb{R}^{d \times n_{\max}}$, we define the positional encoding layer by treating $\theta_{\mathsf{PE}}$ as a mapping from $[n_{\max}]$ to $\mathbb{R}^d$, so that for any $n \in [n_{\max}]$, the positional embedding is $\theta_{\mathsf{PE}}(n)$.

**Output Layer:**   Given output layer parameters $\theta_{\mathsf{OUTPUT}} \in \mathbb{R}^{|\mathcal{V}| \times d}$, we define the output layer $\mathsf{OUTPUT}_{\theta_{\mathsf{OUTPUT}}} : \mathbb{R}^d \to \mathcal{V}$ as $\mathsf{OUTPUT}_{\theta_{\mathsf{OUTPUT}}}(h) = \mathrm{softmax}(\theta_{\mathsf{OUTPUT}} h)$ for any $h \in \mathbb{R}^d$.

A.2   FINITE PRECISION MODELING

We now provide formal definitions for *floating-point numbers* and *rounding* operations. Recall that $\phi(a) = \sum_{i=1}^k 2^{k-i} a_i$ represents the decimal value of a binary string $a \in \{0, 1\}^k$ for any $k \in \mathbb{N}^+$.

**Definition A.1** (Floating-point Representation)**.** *Let $e$ denote the number of exponent bits and $s$ the number of significand bits. An $(e+2s+1)$-bit binary string $a = (a_1, a_2, \ldots a_{e+2s+1}) \in \{0, 1\}^{e+2s+1}$*

---

[2]For simplicity, we omit LayerNorm (Ba et al., 2016) from the standard transformer architecture. Our expressiveness analysis can be extended to transformers with LayerNorm.

---

**Algorithm 2** Embedding Transformer

---

**Require:** Core parameters $\theta_{\text{core}} = (\theta_{\text{PE}}, \{\theta_{\text{ATTN}}^{(l)}, \theta_{\text{FF}}^{(l)}\}_{l=0}^{L-1})$, input content embeddings $e = (e_1, \ldots, e_n) \in \mathbb{R}^{nd}$.

**Ensure:** Output embedding $h \in \mathbb{R}^d$, where $h = \text{EmbTrans}_{(\theta_{\text{PE}}, \{\theta_{\text{ATTN}}^{(l)}, \theta_{\text{FF}}^{(l)}\})}(e)$.

1: $h_i^{(0)} \leftarrow e_i + \theta_{\text{PE}}(i), \forall i \in [n]$
2: **for** $l = 0, \ldots, L-1$ **do**
3: $\quad (h_1^{(l+0.5)}, \ldots, h_n^{(l+0.5)}) \leftarrow (h_1^{(l)}, \ldots, h_n^{(l)}) + \text{ATTN}_{\theta_{\text{ATTN}}^{(l)}}(h_1^{(l)}, \ldots, h_n^{(l)})$
4: $\quad h_i^{(l+1)} \leftarrow h_i^{(l+0.5)} + \text{FF}_{\theta_{\text{FF}}^{(l)}}(h_i^{(l+0.5)}), \forall i \in [n]$
5: **end for**
6: **return** $h_n^{(L)}$

---

**Algorithm 3** Decoder-only Transformer, $\text{TF}_\theta$ and $p_\theta$

---

**Require:** Transformer parameters $\theta = (\theta_{\text{PE}}, \theta_{\text{TE}}, \theta_{\text{OUTPUT}}, \{\theta_{\text{ATTN}}^{(l)}, \theta_{\text{FF}}^{(l)}\}_{l=0}^{L-1})$ and input tokens $x = (x_1, \ldots, x_n) \in \mathcal{V}^n$.

**Ensure:** Output distribution $p_\theta(\cdot \mid x_1, \ldots, x_i)$ for all $i \in [n]$ and output token $\text{TF}_\theta(x)$.
1: $e_i \leftarrow \theta_{\text{TE}}(x_i), \forall i \in [n]$
2: $h_n \leftarrow \text{EmbTrans}_{(\theta_{\text{PE}}, \{\theta_{\text{ATTN}}^{(l)}, \theta_{\text{FF}}^{(l)}\}_{l=0}^{L-1})}(e_1, \ldots, e_n)$
3: $p_\theta(\cdot \mid x_1, \ldots, x_n) \leftarrow \text{OUTPUT}_{\theta_{\text{OUTPUT}}}(h_n)$
4: $\text{TF}_\theta(x) \leftarrow \arg\max_y p_\theta(y \mid x_1, \ldots, x_n)$.

---

*represents a floating-point number $\phi_{e,s}(a) = \text{sign}(a) \cdot 2^{\text{exponent}(a)} \cdot \text{significand}(a)$ with $e$-bit exponent and $2s$-bit precision, where the sign is $\text{sign}(a) = 2a_1 - 1$, the significand is $\text{significand}(a) = 2^{-s}\phi(a_{2:2s+1})$, and the exponent is $\text{exponent}(a) = \phi(a_{2s+2:2s+e+1}) - 2^{\max(0,e-1)}$. We denote by $\mathbb{F}_{e,s}$ the set of all floating-point numbers representable with $e$-bit exponent and $2s$-bit precision:*
$\mathbb{F}_{e,s} = \{S \cdot 2^{-s+E} \mid -2^{2s}+1 \leq S \leq 2^{2s}-1, -2^{\max(0,e-1)} \leq E \leq 2^e - 1 - 2^{\max(0,e-1)}, E, S \in \mathbb{N}\}$.
*We define $B_{e,s} = \max \mathbb{F}_{e,s}$.*

We use $\psi_{e,s} : \mathbb{F}_{e,s} \to \{0,1\}^{e+2s+1}$ to denote the inverse of $\phi_{e,s}$. When the exponent has more than 0 bits, multiple binary strings can represent the same number in $\mathbb{F}_{e,s}$; we choose $\psi_{e,s}(x)$ as the string $a \in \{0,1\}^{e+2s+1}$ with the smallest $|\text{exponent}(a)|$, which is unique for all non-zero numbers. For 0, we additionally set $\text{sign}(\psi_{e,s}(0)) = 1$.

**Definition A.2** (Correct Rounding). *For any $x \in \mathbb{R}$ and any closed subset $\mathbb{F} \subseteq \mathbb{R}$ containing 0, we define correct rounding $\text{round}(x, \mathbb{F})$ as the element in $\mathbb{F}$ closest to $x$. Ties are broken by selecting the element with smaller absolute value.*

*Specifically, we denote rounding with $e$-bit exponent and $2s$-bit precision as $\text{round}_{e,s}(\cdot) = \text{round}(\cdot, \mathbb{F}_{e,s})$, also written as $[\cdot]_{e,s}$ for convenience. We extend $\text{round}$ and $\text{round}_{e,s}$ to vector inputs by applying rounding coordinate-wise.*

Our floating-point representation simplifies the IEEE 754 Standard (Zuras et al., 2008) by excluding $\infty$ and $-\infty$. When overflow occurs, we round the result to the largest representable number (positive or negative) in $\mathbb{F}_{e,s}$. For unary functions like $\exp(\cdot)$ and binary operations including addition, subtraction, multiplication, and division, we define their rounded versions by rounding their outputs. Division by 0 is treated as producing an incorrect result.

Next, we define finite-precision summation over multiple terms by decomposing it into a sequence of rounded binary additions in a fixed order.[3]

**Definition A.3** (Summation with Iterative Rounding). *For any $s, n \in \mathbb{N}^+$ and vector $x \in \mathbb{R}^n$, we define summation with iterative rounding to $e$-bit exponent and $2s$-bit precision as $\text{sum}_{e,s} :$*

---

[3]Technically, summation could also proceed in a tree-like fashion. This more complex case is left for future work.

$\cup_{n\in\mathbb{N}^+}(\mathbb{F}_{e,s})^n \to \mathbb{F}_{e,s}$, *where for any* $n \in \mathbb{N}^+$ *and* $x \in \mathbb{R}^n$,

$$\mathsf{sum}_{e,s}(x) = \left[\left[\left[[x_1 + x_2]_{e,s} + x_3\right]_{e,s} + \cdots x_{n-1}\right]_{e,s} + x_n\right]_{e,s}.$$

*We further define the following operations:*

- *Finite-precision inner product:* $\langle x, y \rangle_{e,s} = \mathsf{sum}_{e,s}(x \odot y)$;

- *Finite-precision matrix multiplication:* $(A \times_{e,s} B)_{i,j} = \left\langle (A_{i,:})^\top, B_{:,j} \right\rangle_{e,s}$;

- *Finite-precision softmax:* $\mathrm{softmax}_{e,s}(x) = \left[[\exp(x)]_{e,s} / \mathsf{sum}_{e,s}([\exp(x)]_{e,s})\right]_{e,s}$.

Finally, a finite-precision transformer is defined by replacing all infinite-precision operations with their finite-precision counterparts listed above. (See details in Algorithm 6). We provide the detailed definitions of finite-precision transformer layers in Appendix A.3.

## A.3 DETAILS ON FINITE-PRECISION LAYERS

In this section, we provide definitions for the finite-precision versions of different transformer layers. Recall that for $s \in \mathbb{N}^+$, the numbers representable with $2s$-bit significand and $e$-bit exponent form the set $\mathbb{F}_{e,s} = \{S \cdot 2^{-s+E} \mid -2^{2s}+1 \le S \le 2^{2s}-1, -2^{\max(0,e-1)} \le E \le 2^e - 1 - 2^{\max(0,e-1)}, E, S \in \mathbb{N}\}$.

**Self-Attention Mechanism:** Given attention parameters $\theta_{\mathsf{ATTN}} = (W_Q, W_K, W_V, W_O) \in \mathbb{F}_{e,s}^{d\times d} \times \mathbb{F}_{e,s}^{d\times d} \times \mathbb{F}_{e,s}^{d\times d} \times \mathbb{F}_{e,s}^{d\times d}$, we define the causal self-attention layer for decoder-only transformers in Algorithm 4.

---

**Algorithm 4** Finite-Precision Causal Self-Attention, ATTN

---

**Require:** Integers $s \in \mathbb{N}^+$, $e \in \mathbb{N}$, Parameter $\theta_{\mathsf{ATTN}} = (W_Q, W_K, W_V, W_O)$, Input embedding $h = (h_1, \ldots, h_n) \in \mathbb{F}_{e,s}^{nd}$.
**Ensure:** Output embedding $h' = (h'_1, \ldots, h'_n) = \mathsf{ATTN}_{\theta_{\mathsf{ATTN}}}(h_1, \ldots, h_n)$.
1: $q_i = W_Q \times_{e,s} h_i, k_i = W_K \times_{e,s} h_i, v_i = W_V \times_{e,s} h_i, \forall i \in [n]$
2: $s_i = \mathrm{softmax}_{e,s}(\langle q_i, k_1 \rangle_{e,s}, \ldots, \langle q_i, k_i \rangle_{e,s}) \| (0, \ldots, 0)$.    ▷ $n - i$ zeros; Mask for Causal Attention;
3: $h'_i = W_O \times_{e,s} \mathsf{sum}_{e,s}([v_1, \ldots, v_n] \times_{e,s} s_i)$.

---

**Algorithm 5** Finite-Precision Embedding Transformer

---

**Require:** Integers $s \in \mathbb{N}^+$, $e \in \mathbb{N}$; core parameters $(\theta_{\mathsf{PE}}, \{\theta_{\mathsf{ATTN}}^{(l)}, \theta_{\mathsf{FF}}^{(l)}\}_{l=0}^{L-1})$ with entries in $\mathbb{F}_{e,s}$; input content embeddings $e = (e_1, \ldots, e_n) \in \mathbb{F}_{e,s}^{nd}$.
**Ensure:** Output embedding $h \in \mathbb{F}_{e,s}^d$, where $h = \mathrm{EmbTrans}_{(\theta_{\mathsf{PE}}, \{\theta_{\mathsf{ATTN}}^{(l)}, \theta_{\mathsf{FF}}^{(l)}\})}^{e,s}(e)$.

1: $h_i^{(0)} = [e_i + \theta_{\mathsf{PE}}(i)]_{e,s}, \forall i \in [n]$
2: **for** $l = 0, \ldots, L-1$ **do**
3:     $(h_1^{(l+0.5)}, \ldots, h_n^{(l+0.5)}) = \left[(h_1^{(l)}, \ldots, h_n^{(l)}) + \mathsf{ATTN}_{\theta_{\mathsf{ATTN}}^{(l)}}(h_1^{(l)}, \ldots, h_n^{(l)})\right]_{e,s}$
4:     $h_i^{(l+1)} = \left[h_i^{(l+0.5)} + \mathsf{FF}_{\theta_{\mathsf{FF}}^{(l)}}(h_i^{(l+0.5)})\right]_{e,s}, \forall i \in [n]$
5: **end for**
6: **return** $h_n^{(L)}$

---

**Feed-Forward Network:** Given $s \in \mathbb{N}^+$, $e \in \mathbb{N}$, and feed-forward network parameters $\theta_{\mathsf{FF}} = (W_1, b_1, W_2, b_2) \in \mathbb{F}_{e,s}^{d\times d} \times \mathbb{F}_{e,s}^d \times \mathbb{F}_{e,s}^{d\times d} \times \mathbb{F}_{e,s}^d$, we define the feed-forward layer $\mathsf{FF}_{\theta_{\mathsf{FF}}} : \mathbb{F}_{e,s}^d \to \mathbb{F}_{e,s}^d$ as $\mathsf{FF}_{\theta_{\mathsf{FF}}}(h) = \left[W_2 \times_{e,s} \mathsf{relu}([W_1 \times_{e,s} h + b_1]_{e,s}) + b_2\right]_{e,s}$.

**Token Embedding:** Given $s \in \mathbb{N}^+$, $e \in \mathbb{N}$, and token embedding parameters $\theta_{\mathsf{TE}} \in \mathbb{F}_{e,s}^{d \times |\mathcal{V}|}$, we define the token embedding layer by treating $\theta_{\mathsf{TE}}$ as a mapping from $\mathcal{V}$ to $\mathbb{R}^d$, so that for any $x \in \mathcal{V}$, the token embedding is $\theta_{\mathsf{TE}}(x)$.

**Position Encoding:** Given $s \in \mathbb{N}^+$, $e \in \mathbb{N}$, and positional encoding parameters $\theta_{\mathsf{PE}} \in \mathbb{F}_{e,s}^{d \times n_{\max}}$, we define the positional encoding layer by treating $\theta_{\mathsf{PE}}$ as a mapping from $[n_{\max}]$ to $\mathbb{R}^d$, so that for any $n \in [n_{\max}]$, the positional embedding is $\theta_{\mathsf{PE}}(n)$.

**Output Layer:** Given $s \in \mathbb{N}^+$, $e \in \mathbb{N}$, and output layer parameters $\theta_{\mathsf{OUTPUT}} \in \mathbb{F}_{e,s}^{|\mathcal{V}| \times d}$, we define the output layer $\mathsf{OUTPUT}_{\theta_{\mathsf{OUTPUT}}} : \mathbb{F}_{e,s}^d \to \mathcal{V}$ as $\mathsf{OUTPUT}_{\theta_{\mathsf{OUTPUT}}}(h) = \mathrm{softmax}_{e,s}(\theta_{\mathsf{OUTPUT}} \times_{e,s} h)$ for any $h \in \mathbb{F}_{e,s}^d$.

Finally, we define finite-precision decoder-only transformers below.

---

**Algorithm 6** Finite-precision Decoder-only Transformer, $\mathsf{TF}_\theta$ and $p_\theta$

---

**Require:** Integers $s \in \mathbb{N}^+$, $e \in \mathbb{N}$. Transformer parameters $\theta = (\theta_{\mathsf{PE}}, \theta_{\mathsf{TE}}, \theta_{\mathsf{OUTPUT}}, \{\theta_{\mathsf{ATTN}}^{(l)}, \theta_{\mathsf{FF}}^{(l)}\}_{l=0}^{L-1})$ with $2s$-bit precision and $e$-bit exponent. Input tokens $x = (x_1, \ldots, x_n) \in \mathcal{V}^n$.
**Ensure:** Output distribution $p_\theta(\cdot \mid x_1, \ldots, x_i)$ for all $i \in [n]$ and output token $\mathsf{TF}_\theta(x)$.
 1: $e_i = \mathsf{TE}(x_i), \forall i \in [n]$
 2: **for** $i = 1, \ldots, n$ **do**
 3: $\quad h_i = \mathrm{EmbTrans}_{(\theta_{\mathsf{PE}}, \{\theta_{\mathsf{ATTN}}^{(l)}, \theta_{\mathsf{FF}}^{(l)}\}_{l=0}^{L-1})}^{e,s}(e_1, \ldots, e_i)$
 4: $\quad p_\theta(\cdot \mid x_1, \ldots, x_i) = [\mathsf{OUTPUT}_{\theta_{\mathsf{OUTPUT}}}(h_i)]_{e,s}$
 5: **end for**
 6: $\mathsf{TF}_\theta(x) = \arg\max_x p_\theta(x \mid x_1, \ldots, x_n)$.

---

# B  MISSING PROOF FROM SECTION 4

Following Li et al. (2024), we introduce the following notations. We will use the shorthand $\mathbb{F}_s \triangleq \mathbb{F}_{0,s} = \{c \cdot k \cdot 2^{-s} \mid c \in \{-1, 1\}, 0 \leq k \leq 2^{2s} - 1, k \in \mathbb{N}\}$ and rounding operation $[\cdot]_s \triangleq [\cdot]_{0,s}$. We use $1_s$ to denote all-one vectors of length $s$. Similarly we define $\langle \cdot, \cdot \rangle_s$, $\times_s$, and $\mathrm{softmax}_s$. We recall that for any $s \in \mathbb{N}^+$ and integer $0 \leq x \leq 2^s - 1$, we use $\mathrm{bin}_s(x) \in \{0, 1\}^s$ to denote the usual binary encoding of integer $x$ using $s$ binary bits in the sense that $x = \sum_{i=1}^s 2^i(\mathrm{bin}_s(x))_i$ and $\mathrm{sbin}_s(x) \in \{-1, 1\}^s$ to denote the signed binary encoding, which is $2\mathrm{bin}_s(x) - (1, \ldots, 1)$.

We also have the following lemmas from Li et al. (2024) that will be used in our proof. Recall $B_s = \max \mathbb{F}_s = 2^s - 2^{-s}$.

**Lemma B.1** (Lemma E.1 Li et al. (2024)). *For any $s \in \mathbb{N}^+$, it holds that $[\exp(-B_s)]_s = 0$.*

Using the same argument above, we also have Lemma B.2.

**Lemma B.2** (Lemma E.2 Li et al. (2024)). *For any $s \in \mathbb{N}^+$, it holds that $[\exp(B_s)]_s = B_s$.*

Given two vectors $x, y$ of the same length $s$, we use $x^\frown y$ to denote their interleaving, that is, $(x^\frown y)_{2i-1} = x_i, (x^\frown y)_{2i} = y_i$ for all $i \in [s]$.

**Lemma B.3** (Lemma E.3 of Li et al. (2024)). *For any $s \in \mathbb{N}^+$, let $q_i = \mathrm{sbin}_s(i)^\frown 1_s$ and $k_i = B_s \cdot (\mathrm{sbin}_s(i)^\frown(-1_s))$ for all $i \in [2^s - 1]$, it holds that $\left[\exp(\langle q_i, k_j \rangle_s)\right]_s = \mathbf{1}[i = j]$ for all $i, j \in [2^s - 1]$.*

**Theorem B.4** (Finite-precision latent CoT upper bound with binary step embedding). *Without loss of generality, let $n, d$ be powers of two with $n = d \cdot k$ where $d \geq \Omega(\log n)$ and let $L_{\mathrm{bin}} := \log_2 k$. There exists a one-layer ($L = 1$) finite-precision embedding transformer (using $\mathbb{F}_s$, $\mathrm{round}_s$, $\mathrm{sum}_s$, and the finite-precision attention/FFN defined in Appendix A.3) that, after*

$$T = k + \lceil \log_2 d \rceil = \frac{n}{d} + \lceil \log_2 d \rceil$$

*steps of latent CoT, outputs a vector $h_{n+T}^{(L)} \in \mathbb{F}_s^D$ whose first coordinate equals the parity of the input bit string $x \in \{0,1\}^n$, i.e., $x_1 \oplus \cdots \oplus x_n$. Moreover, the required model dimension can be taken to be*

$$D = d + L_{\mathrm{bin}} + 3 = d + \log_2 k + 3.$$

*Hence one can decode by reading the first coordinate and thresholding.*

**Model dimension and coordinate layout.** To facilitate explicit layerwise linear/nonlinear constructions while minimizing dimension, write $L_{\mathrm{bin}} := \log_2 k$ and use a model of dimension

$$D = d + L_{\mathrm{bin}} + 3$$

(written as $\mathbb{F}_s^D$ below). The coordinates are partitioned as follows:

- $\mathsf{G}$-slot (coords $1{:}d$): at input tokens, store the *group one-hot* $e_{\mathsf{grp}(j)}$; at scratchpad positions, store the length-$d$ vector of group parities $p^{(r)}$.
- $\mathsf{B}$-slot (coords $d{+}1{:}d{+}L_{\mathrm{bin}}$): a *signed binary code* of the block/step index. For $t \geq 1$, let $\widetilde{b}(t) := \mathsf{sbin}_{L_{\mathrm{bin}}}(t-1) \in \{-1,+1\}^{L_{\mathrm{bin}}}$ denote the signed-binary encoding of $t$. At input tokens $j$, store $\widetilde{b}(\mathsf{blk}(j))$; at scratchpad position $n{+}r$, store $\widetilde{b}(r)$.
- $\mathsf{X}$-slot (coord $d{+}L_{\mathrm{bin}}{+}1$): the input bit $x_j \in \{0,1\}$.
- $\mathsf{1}$-slot (coord $d{+}L_{\mathrm{bin}}{+}2$): the constant 1.
- $\mathsf{Z}$-slot (coord $d{+}L_{\mathrm{bin}}{+}3$): control bits for stage/reduction level (used only in Stage II).

**Input/position embeddings and latent CoT chaining.** Let the token embedding $\theta_{\mathsf{TE}} : \{0,1\} \to \mathbb{F}_s^D$ and position embedding $\theta_{\mathsf{PE}} : [n{+}T] \to \mathbb{F}_s^D$ be

$$\theta_{\mathsf{TE}}(x_j) = x_j \cdot e_{d+L_{\mathrm{bin}}+1}, \qquad \theta_{\mathsf{PE}}(j) = e_{\mathsf{grp}(j)} + \widetilde{b}(\mathsf{blk}(j)) + e_{d+L_{\mathrm{bin}}+2},$$

where $e_i$ denotes the $i$-th standard basis vector (one-hot within its slot), $\mathsf{grp}(j) = ((j{-}1) \bmod d){+}1$, and $\mathsf{blk}(j) = \lceil j/d \rceil$. For scratchpad position $n{+}r$, set

$$\theta_{\mathsf{PE}}(n{+}r) = \widetilde{b}(r) + e_{d+L_{\mathrm{bin}}+2} + \mathbb{1}[r > k] \cdot e_{d+L_{\mathrm{bin}}+3},$$

i.e., the $\mathsf{Z}$-slot is 0 throughout Stage I ($r \leq k$) and flips to 1 in Stage II ($r > k$). Each step of latent CoT feeds the previous output as the next step's "content embedding": given $h_{n+r-1}^{(L)}$, set $e_{n+r} := h_{n+r-1}^{(L)}$ and

$$h_{n+r}^{(0)} = \left[ e_{n+r} + \theta_{\mathsf{PE}}(n{+}r) \right]_{e,s}.$$

(See the finite-precision Embedding Transformer in Algorithm 5.)

**Network structure (two layers per step, $L = 2$).** Each step consists of one finite-precision attention layer (Algorithm 4) and one finite-precision FFN (Appendix A.3):

$$\mathsf{ATTN:} \quad h^{(0.5)} = \left[ h^{(0)} + \mathsf{ATTN}_{\theta_{\mathsf{ATTN}}^{(1)}}(h^{(0)}) \right]_{e,s},$$

$$\mathsf{FFN:} \quad h^{(1)} = \left[ h^{(0.5)} + \mathsf{FF}_{\theta_{\mathsf{FF}}^{(1)}}(h^{(0.5)}) \right]_{e,s}.$$

The attention only needs to produce the *output vector at the current scratchpad position* $n{+}r$; below we analyze only that position.

STAGE I (BLOCK SCAN, $k = n/d$ STEPS): OBTAINING THE $d$ BITS OF THE CURRENT BLOCK IN PARALLEL

**Inductive invariant.** At the beginning of step $r$ ($r = 0, 1, \ldots, k$), the $\mathsf{G}$-slot of $h_{n+r}^{(0)}$ stores $p^{(r)} \in \{0,1\}^d$, the per-group parities over the first $r$ blocks: $p_g^{(r)} = \bigoplus_{\substack{j \leq rd \\ \mathsf{grp}(j)=g}} x_j$. For $r = 0$, $p^{(0)} = \mathbf{0}$.

**ATTN layer: interleaving-based equality with per-step gating (no $L_{\mathrm{bin}}$-sized accumulation).**
Let $s := L_{\mathrm{bin}}$ and reserve the first $2(s+1)$ coordinates of the query/key space for routing; all other
coordinates of $q$ and $k_j$ are 0 and do not affect the inner product. Define (we omit layer indices)

$$\textbf{Gating pair (index } 0\textbf{):} \quad q_1 = 1, \ k_{j,1} = B_s \cdot (2x_j - 1); \qquad q_2 = 0, \ k_{j,2} = 0.$$

$$\textbf{Equality pairs (indices } t = 1, \ldots, s\textbf{):}$$

$$q_{2t+1} = \widetilde{b}_t(r+1), \ k_{j,2t+1} = B_s \, \widetilde{b}_t(\mathsf{blk}(j)),$$
$$q_{2t+2} = 1, \qquad k_{j,2t+2} = -B_s.$$

Take $v_j = e_{\mathsf{grp}(j)}$. Writing the finite-precision partial sums $a_\ell := \langle q_{:\ell}, k_{j,:\ell} \rangle_s$, the interleaving
analysis of Lemma B.3 (applied to the $s$ equality pairs) implies

$$a_{2(s+1)} = \begin{cases} +B_s, & \mathsf{blk}(j) = r+1 \text{ and } x_j = 1, \\ -B_s, & \text{otherwise.} \end{cases}$$

Consequently $\left[\exp(a_{2(s+1)})\right]_s = B_s$ in the first case and 0 otherwise (Lemma B.1, Lemma B.2).
With saturation in $\mathsf{sum}_s$, the softmax denominator clamps to $B_s$ after the first selected token, so each
selected token receives weight 1 and others 0. Therefore

$$y_r = \sum_{\substack{j:\ \mathsf{blk}(j)=r+1 \\ x_j = 1}} e_{\mathsf{grp}(j)} \in \{0,1\}^d.$$

**FFN layer: coordinatewise XOR update $p^{(r+1)} = p^{(r)} \oplus y_r$.** The FFN acts only at the scratchpad
position (gated by the 1-slot), and implements on the G-slot the coordinatewise version of $a \oplus b = a + b - 2\mathrm{ReLU}(a+b-1)$; other slots output 0. All intermediates lie in $\{0,1,2\}$, and when $s \geq 3$
the rounding after addition and ReLU is exact (see the FFN definition in Appendix A.3). Hence the
invariant holds and $p^{(r+1)}$ is written to the G-slot and carried forward by the latent CoT.

**Slot contents at $n+r+1$ (after each sublayer, Stage I).** Using the chaining rule $e_{n+r+1} = h_{n+r}^{(1)}$
and the position embedding above,

$$h_{n+r+1}^{(0)} = \left[ h_{n+r}^{(1)} + \widetilde{b}(r+1) + e_{d+L_{\mathrm{bin}}+2} \right]_s.$$

Thus at $n+r+1$ we have

$$\mathsf{G\text{-}slot} = p^{(r)}, \quad \mathsf{B\text{-}slot} = \widetilde{b}(r+1), \quad \mathsf{X\text{-}slot} = 0, \quad \mathsf{1\text{-}slot} = 1, \quad \mathsf{Z\text{-}slot} = 0.$$

After the attention sublayer,

$$h_{n+r+1}^{(0.5)} = \left[ h_{n+r+1}^{(0)} + \mathsf{ATTN}(\cdot) \right]_s \quad \Rightarrow \quad \mathsf{G\text{-}slot} = p^{(r)} + y_r, \text{ other slots unchanged.}$$

After the FFN sublayer,

$$h_{n+r+1}^{(1)} = \left[ h_{n+r+1}^{(0.5)} + \mathsf{FF}(\cdot) \right]_s \quad \Rightarrow \quad \mathsf{G\text{-}slot} = p^{(r)} \oplus y_r \ (= p^{(r+1)}), \text{ and all non-G slots are reset to 0.}$$

STAGE II (BINARY REDUCTION, $\lceil \log_2 d \rceil$ STEPS): MERGE $d$ GROUP PARITIES DOWN TO ONE BIT

Suppose at the end of step $k$ we have $p^{(k)} \in \{0,1\}^d$ (in the G-slot). At step $k + \ell$ ($\ell = 0, 1, \ldots, \log_2 d - 1$):

- **ATTN (copy previous scratchpad via interleaving equality):** reserve the first $2(L_{\mathrm{bin}}+1)$
  query/key coordinates; use a *gating pair* with $k_1 = B_s(2Z_j - 1)$ (where $Z_j$ is the Z-slot) to
  exclude all non–Stage-II tokens ($Z_j = 0 \Rightarrow -B_s$), followed by $L_{\mathrm{bin}}$ equality pairs on the B-slot
  bits as in Stage I (target index $k+\ell$). The resulting logit equals $+B_s$ iff the token is exactly the
  previous scratch $n+k+\ell$ and $-B_s$ otherwise, so the attention output copies $p^{(k+\ell)}$ losslessly into
  the current step's input.

- **FFN (in-place pairwise XOR):** on the G-slot, set $p_g^{(k+\ell+1)} = p_{2g-1}^{(k+\ell)} \oplus p_{2g}^{(k+\ell)}$ for $g \leq d/2^{\ell+1}$
  and clear the remaining coordinates; gating is controlled by the Z-slot/position embedding to select
  the correct pairing pattern at each reduction level. As in Stage I, the XOR is exact coordinatewise
  under finite precision.

Finally $(p^{(k+\log_2 d)})_1 = \bigoplus_{j=1}^n x_j$. Read this coordinate as the output (optionally followed by a final
linear map to convert 0/1 into an answer token).

**Explicit embeddings at $n+k+\ell+1$ (Stage II).**   For $\ell = 0, 1, \ldots, \log_2 d - 1$, the scratch token at $n+k+\ell+1$ has

$$
h^{(0)}{}_{n+k+\ell+1} = \left[ h^{(1)}_{n+k+\ell} + \widetilde{b}(k+\ell+1) + e_{d+L_{\mathrm{bin}}+2} + e_{d+L_{\mathrm{bin}}+3} \right]_s,
$$

so its slots are

$$
\mathsf{G\text{-slot}} = p^{(k+\ell)}, \quad \mathsf{B\text{-slot}} = \widetilde{b}(k+\ell+1), \quad \mathsf{1\text{-slot}} = 1, \quad \mathsf{Z\text{-slot}} = 1, \quad \mathsf{X\text{-slot}} = 0.
$$

The ATTN sublayer (copy) keeps the $\mathsf{G\text{-slot}}$ unchanged:

$$
h^{(0.5)}_{n+k+\ell+1} : \quad \mathsf{G\text{-slot}} = p^{(k+\ell)} \quad \text{(all other slots unchanged)}.
$$

The FFN then performs in-place pairwise XOR on the $\mathsf{G\text{-slot}}$ according to level $\ell$ and clears the rest:

$$
h^{(1)}_{n+k+\ell+1} : \quad \mathsf{G\text{-slot}} = p^{(k+\ell+1)}, \quad \mathsf{B, X, 1, Z}\text{ slots} = 0.
$$

**Precision and step count.**   Each step uses one finite-precision attention layer (leveraging the *interleaving equality* so that $[\exp(B_s)]_s = B_s$ for selected tokens and $[\exp(-B_s)]_s = 0$ for all others; with saturation in $\mathrm{sum}_s$ the softmax denominator clamps to $B_s$, yielding exact $1/0$ weights) plus one finite-precision FFN to perform coordinatewise XOR. Thus depth $L = 2$ suffices, and the embedding dimension is $D = d + \log_2 k + 3$. The total number of latent CoT steps is $T = k + \lceil \log_2 d \rceil = \frac{n}{d} + \lceil \log_2 d \rceil$. All steps adhere to the finite-precision definitions and algorithms in Appendix A.3.

## C   MISSING PROOF FROM SECTION 3

**Missing proof of the lower bound**   We first prove Lemma 3.3.

*Proof of Lemma 3.3.*  Let $\mathcal{A} \subseteq [n]$ be the set of positions for $z_A$ and $\mathcal{B} \subseteq [n]$ be the set of positions for $z_B$. We use $W_Q, W_K, Q_V$ to denote the query, key, value matrix at the $h$-th attention head of the first layer. We use $x_i \in \mathcal{V}$ to denote the token at position $i$, and use $y_i \in \mathbb{R}^d$ to denote the embedding vector before the attention layer, $y_i' \in \mathbb{R}^d$ to denote the embedding vector after the attention layer.

Consider the following communication protocol. For $r = 1, 2, \ldots, R$, suppose Charlie holds the CoT tokens $x_{n+1}, \ldots, x_{n+r-1}$ at the beginning of round $r$ (Charlie holds an empty string at round 1) and take $z_{C,1} = x_{n+1}, \ldots, z_{C,r-1} = x_{n+r-1}$. At round $r$, the transcript $\Pi_{A,r}$ of Alice is computed as

$$
\Pi_{A,r} := \sum_{j \in \mathcal{A}} \exp(y_{n+r-1}^\top W_Q^\top W_K y_j) W_V y_j || \sum_{j \in \mathcal{A}} \exp(y_{n+r-1}^\top W_Q^\top W_K y_j).
$$

We note the transcript depends only on $\{y_j\}_{j \in \mathcal{A}}$ and $y_{n+r-1}^{(0)}$, which is determined by Alice's input $z_A$ and $z_{C,r-1} = x_{n+r-1}$ (wlog. we assume $x_n$ is a dummy token). The total number of bits are at most $|\Pi_{A,r}| \le (d+1) \cdot p \le 2dp$. Similarly, Bob sends

$$
\Pi_{B,r} := \sum_{j \in \mathcal{B}} \exp(y_{n+r-1}^\top W_Q^\top W_K y_j) W_V y_j || \sum_{j \in \mathcal{B}} \exp(y_{n+r-1}^\top W_Q^\top W_K y_j).
$$

Based on $\Pi_{A,r}$ and $\Pi_{B,r}$, Charlie is able to compute

$$y'_{n+r-1} = \frac{\begin{aligned}&\sum_{j \in \mathcal{A}} \exp(y_{n+r-1}^{\top} W_Q^{\top} W_K y_j) W_V y_j \\ &+ \sum_{j \in \mathcal{B}} \exp(y_{n+r-1}^{\top} W_Q^{\top} W_K y_j) W_V y_j \\ &+ \sum_{j=n}^{n+r-1} \exp(y_{n+r-1}^{\top} W_Q^{\top} W_K y_j) W_V y_j\end{aligned}}{\begin{aligned}&\sum_{j \in \mathcal{A}} \exp(y_{n+r-1}^{\top} W_Q^{\top} W_K y_j) \\ &+ \sum_{j \in \mathcal{B}} \exp(y_{n+r-1}^{\top} W_Q^{\top} W_K y_j) \\ &+ \sum_{j=n}^{n+r-1} \exp(y_{n+r-1}^{\top} W_Q^{\top} W_K y_j)\end{aligned}}$$

Based on $y'_{n+r-1}$, it could compute the next token $x_{n+r} \in \{0,1\}^{\log(|\mathcal{V}|)} = \{0,1\}^p$ and proceed to the next round. This completes the proof.

$\square$

Next we prove Lemma 3.8

*Proof of Lemma 3.8.* By Lemma 3.5, taking $k = 2$ and $m = d$, the communication complexity of any 3-round protocol that solves a single pointer chasing task is at least $\Omega(d)$; by Lemma 3.7, the communication complexity of any 3-round protocol that solves the XOR pointer chasing task with advatanve $\frac{1}{2} + 2^{-n/2d}$ is at least $(n/2d) \cdot \Omega(d) = \Omega(n)$. $\square$

We then prove Lemma 3.9.

*Proof of Lemma 3.9.* Given any $R$ round laconic communication protocol, consider the following 2-round communication protocol for XOR pointer chasing. Alice and Bob first sample $Rp$ random bits $z_1, \dots, z_R \in \{0,1\}^p$ using public randomness. At round 1 (Alice's turn), Alice determines $\Pi_{A,1}, \dots, \Pi_{A,R} \in \{0,1\}^{2dp}$ based on its input and $z_1, \dots, z_R$ according to the laconic protocol and sends them to Bob; Bob also determines $\Pi_{B,1}, \dots, \Pi_{B,R} \in \{0,1\}^{2dp}$ based on its input and $z_1, \dots, z_R$. Now given $\{\Pi_{A,r}\}_{r \in [R]}$ and $\{\Pi_{B,r}\}_{r \in [R]}$, Bob could check whether $z_1, \dots, z_R$ are correct under the laconic communication protocol – note the correctness of $z_r$ depends only on $z_1, \dots, z_{r-1}$ and $\Pi_{A,r}, \Pi_{B,r}$. At round 2, if all $\{z_r\}_{r \in [R]}$ are correct, then Bob returns Charlie's output in the laconic protocol, otherwise, if some of $\{z_r\}_{r \in [R]}$ are incorrect, then Bob randomly returns $0/1$ to Alice.

The above communication protocol proceeds in 2 rounds and its communication complexity equals $R \cdot 2dp + 1$. To analysis its advatange, we note that Alice/Bob correctly guess $z_1, \dots z_R$ with probability $2^{-Rp}$, and whenever $\{z_r\}_{r \in [R]}$ are correct, Alice is able to output the correct answer due to the correctness of laconic protocol. When the guess is incorrect, Bob could spot the error so the output is random. Hence, the overall advantage equals $\frac{1}{2} \cdot (1 - 2^{-Rp}) + 2^{-Rp} = \frac{1}{2} + 2^{-Rp-1}$. By Lemma 3.8, we must have $R = \Omega(n/dp)$ $\square$

**Construction of Latent CoT** We next prove the second part of Theorem 3.1, in particular, we prove that there exists a 1-layer Transformer, when equiping with latent CoT, it could solve the XOR pointer chasing task (Definition 3.6) in $O(n/d^2+1)$ steps.

*Proof of Theorem 3.1, Part 2.* Given $n/2d$ independent 2-step pointer chasing instances (PC$_2$), where each instance $i \in [n/2d]$ is defined by two functions $f_{A,i} : [d] \to [d]$ and $f_{B,i} : [d] \to [d]$.

Our goal is to compute the total arithmetic sum $\sum_{i=1}^{n/2d} v_i$ where $v_i = (f_{B,i} \circ f_{A,i})^{(2)}(1)$ is the composition outcome of the $i$-th function.

Consider a 1-layer Transformer with model dimension $d$ and $H = O(d)$ attention heads. We assume these $n/2d$ functions are encoded by prompt tokens $x_1, \ldots, x_n$ and each token exactly encodes one key-value pair, i.e., $(i, x, f_{A,i}(x))$ or $(i, x, f_{B,i}(x))$ ($i \in [n/2d], x \in [d]$).

The main idea is to use the $d$-dimensional latent vector to run $H = O(d)$ compositions in parallel, with each of the $H$ heads managing one composition. To this end, we group the $n/2d$ function pairs into $B = \lceil n/2dH \rceil$ batches. For $b \in [B]$, the $b$-th batch contains $H$ pairs, i.e., $\{(f_{A,i}, f_{B,i})\}_{i=(b-1)H+h, h \in [H]}$.

The latent CoT would proceed for $B$ macro-steps, where each macro-step $b \in [B]$ computes $H$ values $\{v_i\}_{i \in (b-1)H+h, h \in [H]}$ for the $b$-th batch and adds them to the cumulative sums. Each macro-step futher requires 4 latent CoT steps for computing the composition, so we need $n_{\text{lcot}} = 4B$ latent steps in total. For any $i \in [n_{\text{lcot}}]$, we use $e_{n+i}$ to denote the input embedding vector for position $n + i$, by definition, it is also the output of position $n + i - 1$ (up to position encoding). For any $b \in [B], r \in [4]$, it is convenient to think of the embedding vector $e_{n+4(b-1)+r}$ consisting of three parts. The first part contains $H$ dimension and stores the cumulative sum $e_{n+4(b-1)+r,1} = (S_1^{(b-1)}, \ldots, S_H^{(b-1)}) \in \mathbb{R}^H$, where $S_h^{(b-1)} = \sum_{i=b'H+h, 1 \leq b' \leq b-1} v_i$ (when $b = 1$, $S_h^{(0)} = 0$). The first part is fixed during each macro step $b$. The second part $e_{n+4(b-1)+r,2}$ contains $H$ dimension and it is used to perform composition within the $b$-th batch during $b$-th macro steps. The third part has $O(H)$ dimention and contains positional encoding and some working information.

At each macro step $b$, our goal is to prove that, the 1-layer transformer would equip with the vector

$$e_{n+4(b-1)+1} = (e_{n+4(b-1)+1,1}, e_{n+4(b-1)+1,2}, e_{n+4(b-1)+1,3}) = (S_h^{(b-1)}, \vec{1}, \mathsf{PE}_{n+4(b-1)+1}).$$

Here $\mathsf{PE}_{n+4(b-1)+1}$ is the position encoding at position $\mathsf{PE}_{n+4(b-1)+1}$. The claim holds trivially for $b = 0$ and we would prove by induction. Suppose the claim holds up to $b$, it suffices to prove that, for each step $4(b-1)+r, r \in [4]$, the transformer would compute and output the following vector at position $4(b-1)+r$,

$$e_{n+4(b-1)+1} = (e_{n+4(b-1)+r,1}, e_{n+4(b-1)+r,2}, e_{n+4(b-1)+r,3}) = (S_h^{(b-1)}, (g_i^{(r)}(1))_{i \in [(b-1)H+1:bH]}, \vec{0}).$$

here we define

$$g_i^{(1)}(1) = f_{A,i}(1), \qquad g_i^{(2)}(1) = f_{B,i}(f_{A,i}(1)),$$

$$g_i^{(3)}(1) = f_{A,i}(f_{B,i}(f_{A,i}(1))), \quad g_i^{(4)}(1) = f_{B,i}(f_{A,i}(f_{B,i}(f_{A,i}(1))))$$

for notation convenience. In another word, it performs one step of composition for each function in batch $b$ at each latent CoT step.

To this end, we construct the key, query matrix as follows. The key matrices are identical for each head $h \in [H]$, and it is constructed such that, for each position in the prompt $[n]$, the key value equals $\alpha_{A,i,x} \in \mathbb{R}^{O(\log(d))}$ or $\alpha_{B,i,x} \in \mathbb{R}^{O(\log(d))}$ for $i \in [n/2d]$ and $x \in [d]$, i.e, it equals some encoding vector for the key value of the function. Here we use $\{\alpha_j\}_{j \in \mathsf{poly}(d)}$ to denote a set of encoding vector such that $\langle \alpha_j, \alpha_{j'} \rangle < 7.5 \log(n)$ for $j \neq j'$, while $\langle \alpha_j, \alpha_{j'} \rangle = 10 \log(n)$ for $j = j'$. We note this is satisfied by random vectors in $O(\log(d))$ dimension. The key value for position greater than $[n]$ is always 0 (so it would not be attended). The query matrices are different for each head $h \in [H]$. At step $n+4(b-1)+r$, the query vector at the $h$-th attention head equals $e_{A,(b-1)H+h, g_{(b-1)H+h}^{(r-1)}(1)}$ if $r$ is odd and $e_{B,(b-1)H+h, g_{(b-1)H+h}^{(r-1)}(1)}$ if $r$ is even. By doing this, we can make sure that at the $r$-th step of $b$-th macro step, the $h$-th attention head attends to the position that contains the value $g_{(b-1)H+h}^{(r)}$. Hence each latent CoT step would perform one step composition for $H$ functions simultanously (we omit the detailed construction for the subsequent MLP layer that implements some simple logics).

In summary, the total number of latent CoT steps is $4B = 4 \cdot \lceil n/2dH \rceil = O(n/d^2 + 1)$, and after $B$ macro-steps, the embedding vector $e_{n+4B}$ would contain the vector $(S_1^{(B)}, \ldots, S_H^B)$ as the first part. It remains to perform one step of summation $S = \sum_{h=1}^H S_h^{(B)}$, which can be done within one MLP layer. This completes the proof. □

# D   LLM USAGE

We have utilized LLMs to refine the writing and proofread mathematical proofs.

