# OpenReview forum: "The Information Bottleneck of Chain-of-Thought and How Latent CoT Overcomes It"
_ICLR.cc/2026/Conference — Submitted to ICLR 2026_

### Official Review · Reviewer_i9Xt · 2025-10-19

**Soundness:** 2
**Presentation:** 1
**Contribution:** 3
**Rating:** 4
**Confidence:** 4

**Summary:**

This paper gives a theoretical explanation of why chain-of-thought traces in large language models are often very long: each reasoning step can output only a single token despite internally performing much richer computation. The authors prove theoretically that these problems require long CoTs to be solved, but much shorter ones when using latent CoT, which allows writing high-dimensional vectors. Additionally, it provides experiments demonstrating that latent CoT indeed removes this information bottleneck.

**Strengths:**

1. The authors identify an inherent drawback of the CoT mechanism--its limited information bandwidth--and provide a theoretical explanation for this limitation. This novel theoretical framing gives new insights into this widely observed empirical phenomenon.

2. The authors derive mathematically rigorous bounds on the number of CoT steps required to solve two classic problems--pointer chasing and parity--providing a clear comparison between single-token CoT and latent CoT.

3. The authors introduce a very interesting connection between the dot-by-dot CoT (Pfau et al., 2024), the single-token CoT and latent CoT, framing them as points along a continuum defined by the amount of information transmitted per reasoning step.

4. The paper presents experimental results that validate the theoretical claims.

**Weaknesses:**

1. In the current form, the main weakness of this paper is its presentation. I think the writing could be greatly improved and the paper could be organized better to improve its clarity. For instance, a lot of space is dedicated to section 1.1, which introduces in a lot of detail the contributions of the paper, before introducing many of the formal preliminaries. I think the contributions could be summarized more succinctly in the introduction in a more informal way. This can save space that can be used for adding some informal proof sketches for the main theorems.

2. The observation that single-token CoT is inefficient compared to latent CoT is not new, therefore it seems like the novelty stems from its mathematical framing. However, the paper lacks a precise definition of the information bottleneck, which I think is essential for a theoretical paper of this kind.

3. Another limitation is the relatively limited experimental results. While the synthetic experiments illustrate the information bottleneck, it would help to include experiments on standard reasoning benchmarks, similar to those used in Hao et al. (2024). Such experiments would help demonstrate whether the theoretical insights hold in more realistic settings.

**Questions:**

1. I don't fully understand the paragraph from lines 106-109. Could you elaborate further?

2. How do you formally define the information bottleneck?

3. Hanh and Rofin (2024) prove that the sensitivity of parity using chain-of-thought is constant. How does their finding connect to your theoretical results?

4. Do you think it would be possible to derive similar bounds for problems that can be solved by the dot-by-dot CoT? It would be very interesting to compare not only single-token CoT and latent CoT, but also include dot-by-dot CoT in the analysis.

5. How do you explain the mixed results of latent CoT methods such as COCONUT (Hao et al., 2024) on standard reasoning benchmarks? They observed that performance varies across benchmarks-sometimes exceeding, sometimes matching, and sometimes falling below single-token CoT.



Shibo Hao, Sainbayar Sukhbaatar, DiJia Su, Xian Li, Zhiting Hu, Jason E Weston, Yuandong Tian. 2024. Training Large Language Model to Reason in a Continuous Latent Space.

Michael Hahn and Mark Rofin. 2024. Why are Sensitive Functions Hard for Transformers?. In Proceedings of the 62nd Annual Meeting of the Association for Computational Linguistics (Volume 1: Long Papers), pages 14973–15008, Association for Computational Linguistics.

---

> ### Author Response · Authors · 2025-12-01
> **Response to Reviewer i9Xt**
>
> Thank you for the feedback and for appreciating our novel theoretical framing.
>
> **W1: Presentation and long introduction**
> We agree. We have restructured the paper, shortening Section 1.1 and moving the formal definitions to a new *Preliminaries* section. This improves the flow and allow us to add intuition earlier.
>
> **W2: Novelty and formal definition**
> We agree that there are speculations on the stronger expressive power of latent CoT and this indeed serves as one of the motivations of `Hao et al.`, to the best of knowledge, there is no formal mathematically proof for this conjecture. Our novelty lies in the formal proof on the separation of representation power, which is technical challenging.
> In particular, we are the first to use Communication Complexity and Fourier Analysis to prove *quantitative separations* and *lower bounds* on CoT steps.
>
> We have added a more formal definition for the information bottleneck. Specifically, information bottleneck manifests itself in our proof in the Laconic Communication Model which we defined to capture the information flow in transformer. We have added a Preliminary section, formally defined the Laconic Communication Model and the information bottleneck in it.
>
> **W3: Limited experiments**
> We thank the reviewer for this constructive suggestion. We agree that evaluating on standard reasoning benchmarks is important for demonstrating the practical utility of the proposed method.
> However, we would like to respectfully clarify the primary contribution and scope of our work compared to prior studies like `Hao et al. (2024)`:
> - While `Hao et al. (2024)` and subsequent works have already demonstrated the empirical success of Latent CoT on standard benchmarks, our paper seeks to answer the fundamental question of why standard Token CoT struggles and why Latent CoT is necessary from an information-theoretic perspective.
> - To rigorously verify our "Information Bottleneck" hypothesis, we needed an experimental setting where we could precisely control the information flow. As detailed in Section 5 (Experimental Verification), our Game of Life experiment allows us to tune a specific "bottleneck knob" (the dimension of the vector passed between steps). This enables us to observe the sharp phase transition in performance predicted by our theory—something that would be difficult to isolate in standard reasoning benchmarks due to the varying complexity and ambiguity of natural language tasks.
> - We view our work as complementary to `Hao et al. (2024)`. They established that Latent CoT works in realistic settings; we provide the theoretical grounding for its advantage. Our synthetic experiments are designed to serve as a "controlled laboratory" to validate this theory, rather than to re-establish the general empirical effectiveness of Latent CoT which is already well-documented.
>
> **Q1: Clarify lines 106-109**
> These lines explain how our GoL experimental setup provides a unified view:
> - `bottleneck=0`: No info passes, like "dot-by-dot" CoT.
> - `bottleneck=log(V)`: $O(\log V)$ bits pass, analogous to token CoT.
> - `bottleneck=d`: $O(d)$ bits pass, analogous to latent CoT.
>
> Our experiment (Fig 1) shows that performance collapses as the bottleneck shrinks to token CoT levels, confirming our theory.
>
> **Q3: Hahn and Rofin (2024)**
> Thank you for the reference. The work of `Hahn & Rofin, 2024` conducts Fourier analysis on Transformer but do not consider CoT, whereas the main focus of this work is on the expressive power of tokenized CoT. Our proof for finite precision Transformers uses existing results and tools from circuit complexity, which naturally uses Fourier analysis. We have added more in depth discussion in our paper.
>
>
> **Q4: Bounds for dot-by-dot CoT** This is a good question. For the dot-by-dot CoT, it is equivalent to a multi-step latent CoT with an information bandwidth of 0. Indeed because token CoT has a larger information bottleneck, it might be more efficient: For some problem solvable by dot-by-dot CoT, we might be able to solve it with shorter token CoT. Because dot-by-dot CoT is not actually used in practice, it is not the focus of our paper. But our `bottleneck=0` experiment does cover dot-by-dot CoT.
>
> **Q5: Mixed results of COCONUT** (`Hao et al., 2024`)
> This is a key insight. Our paper provides a theoretical explanation for these mixed results.
> - For tasks with **low state-passing needs** (e.g., simple reasoning), the $O(\log|\mathcal{V}|)$ bottleneck of token CoT is *sufficient*. Latent CoT adds optimization difficulty for no benefit.
> - For tasks with **high state-passing needs** (like our PARITY or GoL), the $O(\log|\mathcal{V}|)$ bottleneck is *insufficient*. Here, latent CoT shows a massive performance gain, as predicted by our theory.
>
> We have added this crucial discussion to our new Related Work section.

---

### Official Review · Reviewer_M3im · 2025-10-19

**Soundness:** 3
**Presentation:** 4
**Contribution:** 3
**Rating:** 8
**Confidence:** 4

**Summary:**

The authors argue that CoT suffers from an information bottleneck, as each reasoning token requires that the model compress its progress into a single token, which for a vocabulary of size $|\mathcal V|$ carries only $O(\log |\mathcal V|)$ bits of information. In contrast, latent CoT with an embedding dimension of $d$ (and constant precision) carries $\Theta(d)$ bits of information per CoT step. Using a communication complexity argument, the authors show how this leads to a separation in the number of CoT rounds needed to solve parity and pointer chasing (for constant-depth transformers). Using a bounded-dimension MLP layer to simulate varying information bottlenecks, the authors show in experiments simulating Conway's game of life that as the problem scales, larger information bottleneck dimensions are required to roll out 10 game of life steps correctly.

**Strengths:**

1. Proving lower bounds on the number of reasoning tokens required to solve a problem is a great contribution. A timely and important problem.
2. The proof techniques, especially applying Fourier analysis, are a nice contribution.
3. The paper is clearly written and presented.

**Weaknesses:**

1. The connection between the theory and the experiments is very loose. Conway's game of life comes out of nowhere, and the experiments do not actually use the vocabulary size as an information bottleneck, just a small intermediate dimension.


### Suggestions
- Figure 1 is quite hard to parse visually as there is no visual pattern indicating which lines have larger or smaller bottlenecks. Using a consistent palette would help (e.g., darker or redder = larger $d$), perhaps in combination with a non-color feature (thicker line = larger). The colors are also inconsistent between $n$s, which compounds the issue
- Some important related work is missing:
   - This paper is very related to the "token complexity hypothesis": How Well do LLMs Compress Their Own Chain-of-Thought? A Token Complexity Approach, https://arxiv.org/abs/2503.01141. Also related, the "Sample optimal length" from ShorterBetter: Guiding Reasoning Models to Find Optimal Inference Length for Efficient Reasoning, https://arxiv.org/abs/2504.21370. This paper can be seen as providing theoretical backing for these empirical results
   - For the idea of information bottlenecks in attention, which is mentioned a couple of times: Lost in Transmission: When and Why LLMs Fail to Reason Globally, https://arxiv.org/abs/2505.08140


### Overall assessment
The theoretical analysis is strong and novel enough to carry the paper, despite the weak connection to the experiments. I think the paper would be substantially better with experiments on the analyzed problems rather than the game of life, however.

**Questions:**

1. Why not parity and pointer-chasing experiments to align with the theory?

---

> ### Comment · Reviewer_M3im · 2025-11-24
>
> I recently discovered the *most* related paper that really needs to be discussed in relation to this work:
>
> Bavandpour, A.A., Huang, X., Rofin, M. &amp; Hahn, M.. (2025). Lower Bounds for Chain-of-Thought Reasoning in Hard-Attention Transformers. Proceedings of the 42nd International Conference on Machine Learning, in PMLR 267:3274-3306. Available from https://proceedings.mlr.press/v267/bavandpour25a.html.
>
>
> This paper shows CoT step lower bounds for parity (as in this work), multiplication, median, and DAG reachability for UHATs (unique hard attention transformers).

---

> > ### Author Response · Authors · 2025-12-01
> > **Response to Reviewer M3im's comment**
> >
> > Thanks for pointing our the references! We have added the discussion on the relationship between our work and those papers in the new "additional related works" section.

---

> ### Author Response · Authors · 2025-12-01
> **Response to Reviewer M3im**
>
> Thank you for the positive review and for highlighting the strength of our theoretical analysis.
>
> **W1 \& Q1: Loose connection (GoL vs. Parity/Pointer)**
> This is a fair critique.  Our theoretical tasks (Pointer Chasing, Parity) were chosen for their known communication complexity and known hardness for finite-precision transformers, allowing us to derive clean lower bounds. Unfortunately, as explained below, these theoretical results do not extend to
> -bit precision transformers. This is due to a fundamental lack of understanding of the circuit class
> , which is a major open problem in circuit complexity.
>
> When it comes to experiments, we want to verify our finding about information bottleneck hold even for
> -bit precision transformers. Thus we chose Conway's Game of Life (GoL) for the experiment because it is a complex, iterative task, likely hard for log-precision transformers. It perfectly illustrates the need for bandwidth. To simulate one step, the model must pass the entire board state—a large amount of information—to the next step. Therefore, GoL serves as an ideal testbed to demonstrate the practical implications of our theory in a regime where formal proofs are impossible. Additionally, directly validating the finite-precision results is impractical due to the instability of training low-precision models. Thus, GoL offers the most robust way to validate our claims in a standard experimental setting.
>
> **Suggestion 1: Figure 1 is hard to parse**
> We agree. We have updated Figure 1 to use a consistent color palette (e.g., blue-to-red) where color consistently map to the bottleneck dimension.
>
> **Suggestion 2: Missing related work**
> Thank you for these valuable references. We added a Related Work section and cite the "Token Complexity Hypothesis" and "ShorterBetter" papers, positioning our work as providing a formal, theoretical-complexity-based explanation for their empirical findings. We also cited "Lost in Transmission" in our discussion of the multi-layer bottleneck.

---

### Official Review · Reviewer_V67T · 2025-10-21

**Soundness:** 3
**Presentation:** 3
**Contribution:** 2
**Rating:** 6
**Confidence:** 3

**Summary:**

The paper analyzes the computational abilities of chain of thought (CoT). In particular, it analyzes how the discrete and one-token-at-a-time nature of CoT limits its efficiency and requires a large number of model evaluations to solve certain problems. Methodologically, they approach this through the lens of information bottleneck, showing that each CoT step only adds $\log |\mathcal{V}|$ information, where $\mathcal{V}$ is the vocabulary. In contrast, if the model does not have to decode at every time step (such as in latent CoT or regression-based transformers), the amount of information passed can be much larger. With this, the authors showcase two problems (pointer chasing and parity), where transformers require a large number of CoT steps (model evaluations) when using decoding, but latent CoT substantially reduces the number of model evaluations required.
They evaluate their theory by training transformers on the Game of life and controlling how much information the CoT tokens can convey. They find a close correspondence with the theory, as communicating more information starkly improves performance with a limited number of model evaluations.

**Strengths:**

- I believe the paper tackles a relevant problem that is interesting both theoretically as well as practically.
	- For example, in the theoretical section, it considers languages known in theoretical literature to be difficult for transformers
- I also like that the paper applies standard expressivity results on the relationship between sensitivity and circuit complexity to modern LMs.
- The setup is clear, and the experiments make sense.
	- For example, studying the three different regimes (zero, log, full) make sense
	- The experiments are also well-documented
	- The main results are clearly presented, and I think they are a valuable contribution
- I like that the paper uses a known theoretical transformer framework
- Proofs seem sound and ground the contribution

**Weaknesses:**

- I believe `Amiri et al., Lower Bounds for Chain-of-Thought Reasoning in Hard-Attention Transformers, 2025` studies a very similar problem of the lower bounds on the number of CoT steps and solves a seemingly more general problem in some ways. However, it is not discussed anywhere.
- I am not familiar with communication complexity, so I found the discussion and presentation of the results on the lower bounds very hard to follow, as many technical terms were used without being introduced, for example:
	- What is communication complexity at all?
	- What does it mean for a protocol to succeed with some probability?
	- What is randomized communication complexity?
	- What is advantage?
- The concepts about sensitive functions seem to be based on Ryan O'Donnell’s Analysis of Boolean Functions; I believe it should be cited.
	- I do think sensitive functions could be introduced later in the text though, as they don’t fit naturally where they are introduced/they are not used immediately
- The extra page (current submission is only 8 pages long) could be used for introducing communication complexity terms and for more discussion and a conclusion.

**Questions:**

- How is the latent-CoT supervision loss computed when `information_bottleneck` is logarithmic or zero? The one-hot-encoded information would not be of the right size.
- How do you ensure that the tokenizer agrees with the steps of the (latent) CoT?
- Why did you encode game of life in natural language? Wouldn’t it be more targeted to encode it with some specialized syntax?
- Could you comment on the difference between your results and those by Amiri et al.? This would help clarify the paper’s contributions.
- Have you thought about frameworks such as looped/universal transformers and diffusion models? The parallel generation inherent to those models could make problems solvable with fewer model evaluations.
	- In particular, what is the connection between your framework and that of Saunshi et al., Reasoning with latent thoughts and other “latent reasoning” frameworks that perform updates to the residual stream in parallel?
- Are there any practical takeaways for improving CoT? Larger vocabularies? Including latent CoT?

---

> ### Author Response · Authors · 2025-12-01
> **Response to Reviewer V67T**
>
> Thank you for the constructive feedback and for appreciating our theoretical grounding.
>
> **W1 & Q4: Missing related work (Amiri et al.)**
> This is a critical point. We added a discussion about `Amiri et al., Lower Bounds for Chain-of-Thought Reasoning in Hard-Attention Transformers, 2025` Their work focuses on lower bounds for *hard-attention* Trans, which is a different Transformer model than the commonly used soft-attention Transformers. The proof technique is very different, and we shall add a discussion that explains how our proof technique can be used to replica their main Theorem. In particular, hard attention Tranformer could be simulated by AC circuit, so the proof in Section 4 would go through and obtain similar results as Amiri et al. Meantime, `Amiri et al., 2025` uses a completely different approach, which directs extend upon the work of `Hahn, 2020` and proves via ad-hoc switching Lemma developed in that paper.
>
>
> **W2 \& W5: CC terms and 8-page limit**
> We agree. We have revised the paper and used the 9th and 10th page to add a paragraph defining basically terminology of for communication complexity.
>
> **W3 \& W4: O'Donnell citation and Sensitivity**
> Thanks for the suggestions. This is a standard result for Boolean function and we added a citation to O'Donnell's "Analysis of Boolean Functions".
>
> **Q1: Latent loss for bottleneck**
> We added more explaination to clear this confusion. The supervision $\mathcal{L}_{latent}$ is always an L2 loss against the ground-truth one-hot board state. This loss is applied to the output of the `mlp1` module, which is the $2 n^2$-dimensional vector *before* the bottleneck layer. The information bottleneck is within the `mlp2` module afterwards.
>
> **Q2: Tokenizer and latent CoT steps**
> In our Game of Life experiments, the CoT is indeed purely latent. The tokenizer is used only for encoding the initial natural language prompt and decoding the final numerical answer. The intermediate $k=10$ steps consist of continuous latent vector generation rather than discrete token sampling.
>
> To systematically study the impact of bandwidth, we simulate the information constraint of a standard text tokenizer (which allows $\approx \log |\mathcal{V}|$ bits per step) by adjusting the width of the \texttt{info\_bottleneck} layer in these latent steps. This approach provides a \textbf{unified view} that is more general than a specific tokenizer: it allows us to vary the communication bandwidth continuously from $0$ (dot-by-dot CoT) to $\approx \log |\mathcal{V}|$ (standard token CoT) and up to $d_{model}$ (fully latent CoT), thereby isolating the fundamental role of information capacity in the reasoning process.
>
> **Q3: Why natural language for GoL?**
> To demonstrate that this bottleneck is a practical concern for standard LLMs. The model must first parse a natural language prompt before entering its internal, high-bandwidth simulation phase.
>
> **Q5 \& Q6: Looped transformers and Saunshi et al.**
> These are great connections. We added them to our Related Work. Looped transformers could be seen as one way to *implement* a latent CoT. Our work provides a theoretical justification for *why* such latent reasoning (as in `Saunshi et al.`) is necessary for certain tasks.
>
> **Q7: Practical takeaways**
> Yes. (1) The $O(\log|\mathcal{V}|)$ bottleneck is a fundamental limitation for tasks requiring high-bandwidth state transfer. (2) Latent CoT is a theoretically and empirically sound method to overcome this. (3) Simply increasing model depth may not be a panacea for high-sensitivity problems due to finite-precision limits (Sec 4).

---

### Official Review · Reviewer_ZUA8 · 2025-10-30

**Soundness:** 2
**Presentation:** 2
**Contribution:** 2
**Rating:** 2
**Confidence:** 3

**Summary:**

This paper leverages communication complexity to formally analyze the limitations of token-level CoT. The authors establish that token-level CoT faces an information bottleneck on pointer-chasing and parity tasks, proving lower bounds on the required CoT length. In contrast, this paper demonstrates that a latent CoT approach can overcome this bottleneck, requiring a shorter CoT length and thus possessing greater expressiveness. Experimental results on Conway's Game of Life are presented to empirically support the theoretical claims.

**Strengths:**

The paper introduces a novel and valuable perspective by applying communication complexity to formally analyze the theoretical properties of latent CoT vs. standard token-level CoT. This is an interesting and promising theoretical direction for understanding the capabilities and limitations of different reasoning approaches.

**Weaknesses:**

1. This paper appears to be an incomplete draft. It is missing several standard and essential sections, such as Related Work and Conclusion sections. The main text's length (8 pages) also suggests it may not be finalized. For a submission, the paper would need to be presented as a complete work.

2. This paper would be more accessible and self-contained with the inclusion of more preliminaries introduction, particularly the necessary background from communication complexity. Additionally, the problem setups could be clarified: a more detailed description of the tasks (pointer chasing, parity), especially the specific input/output representations for the Transformer is needed for the results to be fully understood.

3. The theoretical claims require more rigorous support. For instance, Theorem 4.1 is presented with only a proof sketch in the main text, and a complete, formal proof appears to be missing from the appendix. All theorems should be supported by complete and verifiable proofs.

4. The experimental validation is currently limited to a single task (Conway's Game of Life). It would be highly beneficial to include experiments on a more diverse set of reasoning tasks. Moreover, there is a notable disconnect between the tasks analyzed theoretically (pointer chasing, parity) and the task used for experiments. Aligning the theoretical and empirical investigations more closely, or clearly justifying the choice of the experimental task in the context of the theory, would substantially strengthen the paper's core argument.

**Questions:**

1. In Theorem 3.1, if one sets $d = \Omega(n^{0.6})$, the bound $n / d^2$ becomes $O(n^{-0.2})$, which approaches zero as $n \to \infty$. Could the authors clarify if this implies that no CoT steps are needed in this regime?

2. The proof of Theorem 3.1 appears to rely on Definition 3.6, which involves $n / 2d$ functions. Does this reliance imply that Theorem 3.1 holds only under the assumption that $d \leq n/2$? Please clarify the conditions and domain for this theorem.

3. The paper considers finite precision Transformers. Could the authors discuss whether these theoretical results are expected to generalize to models using $O(\log n)$-bit precision?

---

> ### Author Response · Authors · 2025-12-01
> **Response to Reviewer ZUA8**
>
> **W1, W2: Missing related works and prelims)**
>
>
>
> We have cited major relevant works in our main paper. But we agree an *Additional Related Work* section is helpful to keep track with the fast growing line of work on this direction, we have added it in our revision. In this new section, we discussed the relation between our work and works including:
>
>
>
> `Bavandpour et al., Lower bounds for chain-of-thought reasoning in hard-attention transformers. , 2025`,
>
>
>
> `Hahn & Rofin, Why are sensitive functions hard for transformers?, 2024`,
>
>
>
> `Lee et al., How well do LLMs compress their own chain-ofthought? a token complexity approach, 2025`,
>
>
>
> `Yi et al., Shorterbetter: Guiding reasoning models to find optimal inference length for efficient reasoning, 2025`,
>
>
>
> `Schnabel et al., Lost in transmission: When and why llms fail to reason globally, 2025`.
>
> We have also added a preliminary section, explaining the basics of communication complexity and the description of the tasks studied.
>
> Finally, regarding the comment on the 8-page length, we respectfully note that the completeness of a work is not defined by filling the maximum allowable page count. We believe our initial submission covered the necessary technical depth, but we have happily utilized the additional space in the revision to incorporate the suggested discussions.
>
> **W3: Rigor of Theorem 4.1**
> We respectfully clarify that Thm 4.1 is fully proved. The main text provides the proof for the lower bound. The (constructive) upper bound for latent CoT ($O(n/d + \log n)$ steps) is formally stated and proved as Theorem C.4 in Appendix C. We stated this right after Thm 4.1. We apologize for the confusion and we have made it more apparent in the main text.
>
> **W4: Gap between theory and experiments**
>
> This is a fair point. Our theoretical tasks (Pointer Chasing, Parity) were chosen for their known communication complexity and known hardness for finite-precision transformers, allowing us to derive clean lower bounds. Unfortunately, as explained below, these theoretical results do not extend to $\log (n)$-bit precision transformers. This is due to a fundamental lack of understanding of the circuit class $\textbf{TC}^0$, which is a major open problem in circuit complexity.
>
> When it comes to experiments, we want to verify our finding about information bottleneck hold even for $\log (n)$-bit precision transformers. Thus we chose Conway's Game of Life (GoL) for the experiment because it is a complex, iterative task, likely hard for log-precision transformers. It perfectly illustrates the *need for bandwidth*. To simulate one step, the model *must* pass the entire $n \times n$ board state—a large amount of information—to the next step. Therefore, GoL serves as an ideal testbed to demonstrate the *practical* implications of our theory in a regime where formal proofs are impossible. Additionally, directly validating the finite-precision results is impractical due to the instability of training low-precision models. Thus, GoL offers the most robust way to validate our claims in a standard experimental setting.
>
>
> **Q1: "$O(n/d^2)$" bound if $d > \sqrt{n}$**
> The bound $O(n/d^2)$ refers to the number of latent CoT steps. If $d > \sqrt{n}$, the bound becomes $O(1)$, not zero. This is precisely our point: with sufficient model dimension, latent CoT can solve the problem in a *constant* number of steps, whereas token CoT still requires $\Omega(n/d)$ steps. In revision, we clarified this by writhing the bound as "$O(n/d^2 + 1)$".
>
> **Q2: Thm 3.1 dependence on Def 3.6**
> Yes, Thm 3.1's lower bound is proven for the "XOR pointer chasing" task (Def 3.6). This task is constructed by XORing $n/2d$ independent instances, hence our lower bound holds only when $d<n/2$, actually when $d \geq n/2$, both latent and vanilla CoT could solve the question in constant steps.
>
> **Q3: Generalizing to $b$-bit precision**
> Our finite-precision results (Sec 4) rely on the model's computation being approximable by low-degree polynomials (Thm 4.2). This property holds for any *fixed* precision $b$. This result do not extend to $\log(n)$ precision Transformer. Indeed, with $\log(n)$ precision, the PARITY can be represented by constant depth Transformer so it can be be captured by low-degree polynomial.

---

### Official Review · Reviewer_VdsG · 2025-10-31

**Soundness:** 3
**Presentation:** 2
**Contribution:** 3
**Rating:** 8
**Confidence:** 3

**Summary:**

The paper theoretically compares transformer with word space CoT against latent space CoT. By looking at specific example tasks, the paper shows that a 1-layer transformer with latent CoT is much faster (at least with a factor of model dimension) than standard CoT. This is done by looking at a specific task (pointer chasing) and providing lower bound for number of tokens in a standard CoT and an upper bound for number of tokens with a latent CoT. Similarly, for a multi layer transformer, as long as the computation precision is finite, the authors show a similar task where there is a gap between CoT and latent CoT (at least with a factor of model dimension). Finally, the paper looks at a special version of solving the simulation of Game of Life. In particular, the settings allows the authors to control the information flow between CoT steps. They show that the task can be solved only if bottleneck is large enough to showcase the importance of this bottleneck.

**Strengths:**

The paper considers several interesting settings. Focusing on the single layer transformers is interesting as there is no room for a transformer to pass additional information such as intermediate vectors through attention. This is extended through a different method and by cleverly utilizing the finite nature of the available computation precision to multi-layer transformers. Also, proposing a practical problem where the effect of this gap can be clearly observed is also interesting.


The paper is written mostly clearly and the authors do a good job of keeping things mostly simplified and postponing the details to the appendix.

**Weaknesses:**

I believe some of the claims (especially some made in the introduction section) are inaccurate. For example, when considering multi-layer trnasformers, it is no longer true that the information that can be passed is bounded by log V. These models generate latent representation in intermediate layers which future tokens can attend to. I strongly suggest to make these claims more accurate.

In Section 4, it would be useful to add a discussion around the effect of number of transformer layers and discuss how it is not possible to asymptotically match the increase of context length using a fixed depth. Similarly, it would be nice to more clearly highlight the effect of fixed precision. Generally, I feel this section could use a bit more discussion to make things more intuitive and clear.

**Questions:**

1. Is it possible to show what is proved for parity task also in practice?

---

> ### Author Response · Authors · 2025-12-01
> **Response to Review VdsG**
>
> Thank you very much for your time and effort in reviewing our paper!
>
> **W1: Inaccurate claims in introduction (multi-layer transformers)**
> Thanks for pointing this out. We have added more explanation on this claim. In particular, we emphasized that that while attention can be think of as an *internal* high-bandwidth channel that pass rich, high-dimensional internal states around, this information is forced to be abruptly compressed into a single token at the final layer. Since the computation within a single forward pass is non-recurrent and limited by the fixed number of layers, the model must rely on the explicit CoT to maintain state for complex multi-step reasoning. The *explicit, autoregressive transcript* (the CoT itself) remains a bottleneck, forcing the model to write its state through the $O(\log|\mathcal{V}|)$-bit narrow channel.
>
> **W2: More intuition for Section 4 (multi-layer)**
> Great suggestion. We added a discussion to Section 4 explaining the interplay between the number of layers $L$ and the context length $n$.
>
> **Q1: Show parity task in practice?**
> The parity task result relies on the assumption of finite-precision transformers. This condition is theoretically necessary because Parity belongs to $\textbf{TC}^0$, the complexity class characterizing constant-depth, log-precision transformers (without CoT). Practically, this setting is analogous to low-bit (e.g., 4-bit) quantized models. However, experimentally verifying our results by training 4-bit models is challenging, as optimization with discrete weights (e.g., via straight-through estimators) is notoriously unstable.
>
> Therefore, we validated the information-bandwidth theory using a controlled study on Conway's Game of Life. This ``state tracking'' problem is likely $\textbf{NC}^1$-complete and thus lies outside of $\textbf{TC}^0$. This distinction is crucial: since $\textbf{TC}^0$ upper-bounds the expressivity of constant-depth Transformers, a task outside this class provably requires sequential reasoning (CoT) and cannot be solved by a parallel shortcut. This ensures that the success/failure of the model depends strictly on the information bandwidth available between CoT steps, allowing us to cleanly isolate the information bottleneck effect.

---

### Meta-Review · Area_Chair_K4FG · 2026-01-07

**Summary:**

The paper introduces the information bottleneck in chain-of-thought (CoT) reasoning, showing that token-level CoT faces fundamental limitations due to its narrow bandwidth compared to latent CoT approaches. While the theoretical framework using computational complexity theory is novel and mathematically sound, the paper suffers from significant gaps between its theoretical claims and experimental validation. The most critical concern across reviews is the disconnect between the theoretically analyzed tasks (pointer chasing and parity) and the experimental validation on Conway's Game of Life. Additionally, the paper initially lacked essential sections (Related Work, Conclusion), had presentation issues for readers unfamiliar with communication complexity, and omitted discussion of highly relevant recent work. Despite the authors' substantial efforts to address these concerns in their rebuttal, fundamental issues regarding practical validation and the applicability of theoretical results to real-world settings remain unresolved.

**Reviewer Concerns:**

## Addressed concerns:

* The authors have added a comprehensive Related Work section addressing previously missing citations.
And the presentation issues have been partially resolved through restructuring the paper, shortening the introduction, and adding a Preliminary section explaining communication complexity concepts.
* The authors clarified their claims about multi-layer transformers and the nature of information bottlenecks.
* Questions about experimental setup (particularly regarding the Game of Life experiments) were thoroughly addressed.
* Visual presentation issues in Figure 1 were acknowledged and improved.
## Outstanding concerns:

* The fundamental disconnect between theory and experiments remains problematic. While the authors provide justification for using Game of Life instead of parity/pointer chasing tasks, this gap weakens the paper's overall contribution. Their argument that finite-precision experiments are "impractical due to instability of training low-precision models" is not fully convincing as an explanation for the lack of alignment between theoretical and experimental tasks.
* The paper still lacks validation on standard reasoning benchmarks that would demonstrate the practical relevance of their theoretical findings.
* Despite added explanations, the applicability of their finite-precision theoretical results to standard floating-point precision models used in practice remains unclear.

**Reviewer Scores:**

* **Reviewer VdsG**: Initially rated 8. Their concerns about multi-layer transformer claims and section 4 discussion were well-addressed. Likely would maintain a high score but might have lingering concerns about experimental validation.
* **Reviewer ZUA8**: Initially rated 2. While the authors addressed formatting issues by adding missing sections and improved presentation, this reviewer's fundamental concerns about rigor and the theory-experiment gap were only partially resolved. Likely would maintain the strong rejection after discussion.
* **Reviewer V67T**: Initially rated 6. The authors adequately addressed their concerns about related work and terminology. Likely score after discussion: 6 or 8, though practical validation concerns might temper enthusiasm.
* **Reviewer M3im**: Initially rated 8. Despite strong appreciation for the theoretical contributions, this reviewer's primary concern about the theory-experiment disconnect was not fully resolved. Likely score after discussion: 6 or 8.
* **Reviewer i9Xt**: Initially rated 4. The authors addressed presentation issues and added formal definitions for the information bottleneck. However, concerns about limited experimental validation on standard benchmarks remain partially unaddressed. Likely score after discussion: 4.

---

### Decision · Program_Chairs · 2026-01-26

Reject